# Ground-Coupled Natural Circulating Devices (Thermosiphons): A Review of Modeling, Experimental and Development Studies

**Messaoud Badache [1],*, Zine Aidoun [1], Parham Eslami-Nejad [1] and Daniela Blessent [2]** 

1   CanmetENERGY Natural Resources Canada, 1615 Lionel Boulet Blvd., P.O.Box 4800, Varennes, QC J3X1S6, Canada; Zine.aidoun@canada.ca (Z.A.); Parham.eslaminejad@canada.ca (P.E.-N.)
2   Department of environmental engineering, Universidad de Medellín, Medellín 50026, Colombia; dblessent@udem.edu.co
*   Correspondence: Messaoud.badache@canada.ca; Tel.: +1-450-652-6894

**Abstract:** Compared to conventional ground heat exchangers that require a separate pump or other mechanical devices to circulate the heat transfer fluid, ground coupled thermosiphons or naturally circulating ground heat exchangers do not require additional equipment for fluid circulation in the loop. This might lead to a better overall efficiency and much simpler operation. This paper provides a review of the current published literature on the different types of existing ground coupled thermosiphons for use in applications requiring moderate and low temperatures. Effort has been focused on their classification according to type, configurations, major designs, and chronological year of apparition. Important technological findings and characteristics are provided in summary tables. Advances are identified in terms of the latest device developments and innovative concepts of thermosiphon technology used for the heat transfer to and from the soil. Applications are presented in a novel, well-defined classification in which major ground coupled thermosiphon applications are categorized in terms of medium and low temperature technologies. Finally, performance evaluation is meticulously discussed in terms of modeling, simulations, parametric, and experimental studies.

**Keywords:** thermosiphon; heat pipe; ground-coupled natural circulating devices; modeling and experimental

## 1. Introduction

In recent years, the interest of the scientific community in geothermal equipment, such as technologies capable of taking advantage of renewable energy extraction, has significantly increased to become a top priority. One of the main drivers of this interest is a need to render geothermal equipment economically accessible and reliable for the preservation of the environment. Given the current context of climate change, marked with the increases in global temperatures and energy consumption resulting from industrial activities, energy efficiency has become an urgent economic and environmental necessity.

Geothermal heat is an almost infinite renewable energy, with a vast potential and a large thermal inventory [1]. It can be classified into two main usable categories: deep and shallow. Shallow geothermal energy refers to the thermal potential of the near-surface zone (from 1 m to 400 m). Deep thermal potential refers to the heat available in the zone beyond 400 m deep [2]. Heat from the ground can be extracted in two ways: by heat transfer from an aquifer or from the rocks surrounding the well, with or without fluid extraction. Heat extraction can be active, with the use of mechanical means such as circulation pumps and Ground Coupled Heat Pumps (GCHP); passive, with the use of naturally circulating devices such as thermosiphons; or combined (passive and active). The heat transfer fluid

may be air, a water glycol solution, or any suitable refrigerant. All these geothermal systems have one thing in common: balance between heat extraction and natural recharge has to be known beforehand and must be satisfied in order to maintain the renewability of the source. Franco and Vaccaro [3] summarized the main strategies and practices for geothermal resource utilization.

Several types of ground heat exchangers have been proposed and used in order to transfer heat to and/or from the ground [4,5]. The most frequent are earth-air heat exchangers, naturally circulating Ground Coupled Heat Exchangers (GCHE), and actively operated (i.e., with the use of a circulation pump)borehole heat exchangers (BHEs).Compared to conventional mechanical refrigeration systems, the use of naturally circulating GCHE avoids the use of moving parts and can transfer large quantities of heat efficiently over large distances. System operation can be performed at a constant temperature, without requiring any external electricity input. In addition, these systems can be used with a large variety of natural working fluids (e.g., $CO_2$, ammonia, pentane, and propane) and their operation is free from contamination as no lubricant oil is circulated inside the system.

Heat extraction and transfer from the ground by means of natural circulating GCHEs is not a new idea. These devices have been widely used since the late 1950s [6], in transportation infrastructure (highways, bridges, railways, airport runways), civil buildings, oil pipelines, transmission lines towers, and dams. They have also become a powerful technology used to maintain the stability of slab-on-grade foundations in permafrost regions [7]. Most of these systems have been installed in North America, Europe, Russia, and China [8].

To date, several hundred scientific papers have been published on natural circulating GCHE, discussing various subjects ranging from design methodologies and applications to detailed heat transfer models. An interesting classification of the naturally circulating GCHE devices is proposed herein, based on the work of [9,10]. As depicted in Figure 1, four major classes are identified:

- Open, single phase, naturally circulating devices (red colour);
- Closed, single phase, naturally circulating devices (green colour);
- Wickless, closed, two-phase, naturally circulating devices (blue colour);
- Closed, two phase, devices with a Wick structure (white colour).

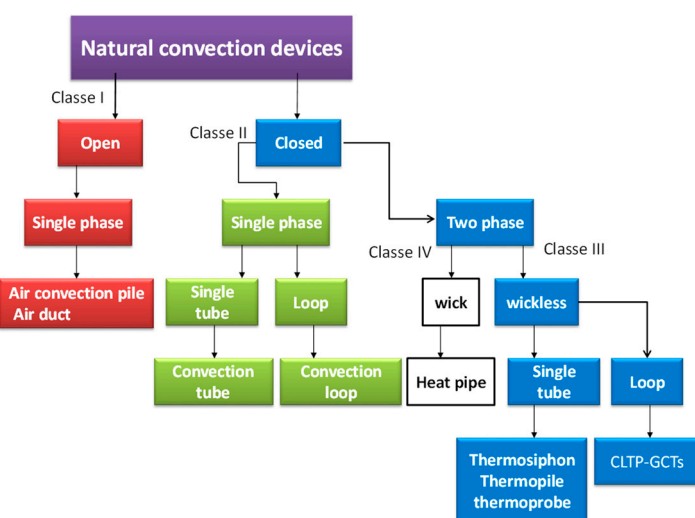

**Figure 1.** Classification of naturally circulating devices for heat extraction from the ground.

Based on this classification, the specific interest of the present investigation will be devoted to wickless single tube and closed loop, two-phase ground coupled naturally circulating devices, known herein as ''Ground Coupled Thermosiphons" (GCT). The paper will specifically provide a review based on the currently published literature about the different types of existing Single Tube Ground CoupledThermosiphons (ST-GCTs) and Closed Loop, two-phase, Ground Coupled Thermosiphons

(CL-GCTs), which are used in a variety of applications requiring moderate and low temperature sources. A brief historical overview of the technology is given. Advances are identified in terms of the latest device developments. The different applications are presented in a novel, well-defined classification. Performance evaluation is meticulously discussed in terms of mathematical modelling, simulations, and parametric studies, as well as experiments.

### 1.1. Operation of GCTs

An ST-GCT system (Figure 2a) is a natural two-phase circulating device that extracts heat from the ground (evaporator) and discharges it at the condenser (atmosphere in some cases). A typical unit is a vertical tube or vessel, closed at both ends, and charged with two-phase working fluid. The vapour phase of the working fluid occupies most space in the pipe, while the liquid phase occupies a small part of the total volume. A heat pipe is a device similar to the GCT, but differs from it in that it usually contains an internal wick for moving the condensate by means of capillary action. In an ST-GCT, the condensate flows by gravity alone. Typically, an ST-GCT is installed with a portion of the pipe above ground and the remainder is buried. The above ground portion of an ST-GCT forms the condenser and the portion below the ground is the evaporator. The portion between the evaporator and the condenser is known as the adiabatic section. In some cases, the underground evaporator part is inclined 3 to 10% from horizontal [11]. When the liquid phase in the evaporator absorbs heat from the ground, it evaporates and flows upward through the pipe to reach the condenser. When this working fluid is cooled below the saturation temperature of the vapor, the vapor condenses and releases its latent heat of vaporization. The condensate in the upper portion of the ST-GCT flows by gravity to the lower portion of the ST-GCT. The cycle continues as long as the above-ground portion is colder than the below ground portion of the ST-GCT. When the temperature in the above ground portion is greater than the ground temperature, heat transfer through the device ceases [12]. A CL-GCT works in the same manner, except that the evaporator and the condenser are connected by two tubes, the riser and the down-comer (Figure 2b). In this case, the ascending vapor and the descending liquid generally do not interact. A particular case of CL-GCTs is the geothermal convector (GC), proposed by Carotenuto et al. [13] for heat extraction from geothermal aquifers.

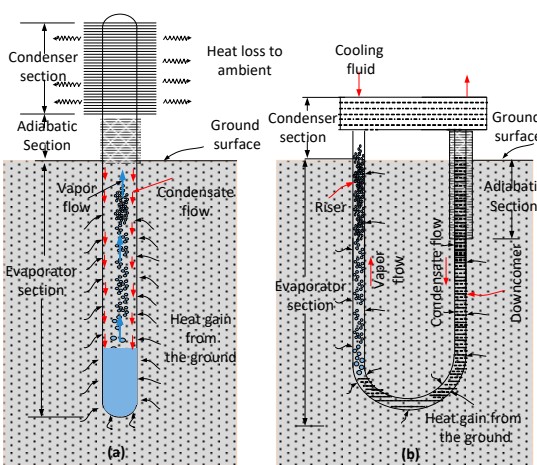

**Figure 2.** Schematic views of (**a**) ST-GCT and (**b**) CL-GCT.

Besides a large list of advantages, GCTs have their limitations. The main disadvantage is that traditional GCT technology does not work in cooling mode, but only in heating mode, because the condenser only works in one direction [14]. In addition, the first costs, especially of the drilling, are very high. However, this weakness is not specific to GCTs, but is the case for every form of vertical ground source heat exchanger.

Depending on the circulation mode, GCT systems can be classified as passive, active, and hybrid systems [8]. A schematic view of active and hybrid systems is shown in Figure 3. As explained

above, passive systems operate only when the above-ground air temperature is lower than the ground temperature. Active systems incorporate a heat exchanger into the condenser, which allows the thermosiphon to connect to a heat pump. This allows the system to operate independently of the ambient air temperature. A hybrid system combines passive and active systems. This means that it can operate as a passive system when the ambient temperature is sufficiently low and as an active system when the air temperature becomes too warm to condense the refrigerant, or when other system conditions necessitate additional ground cooling. According to Wagner [15], hybrid systems can either be used to increase the rate of ground freezing instantly after the installation of thermosiphons, particularly in the summer, or to offer a backup freezing source for eventual situations, when the thaw depth exceeds expectations, when rapid cooling is required for construction scheduling, orwhenthere is a risk of unexpected heat peaks. Originally, hybrid systems were used when more cooling was required to prevent permafrost thawing beneath civil constructions [11], but later, this practice was extended to other types of applications [16,17], with certain modifications such as the creation of artificial frozen barriers using hybrid thermosiphons.

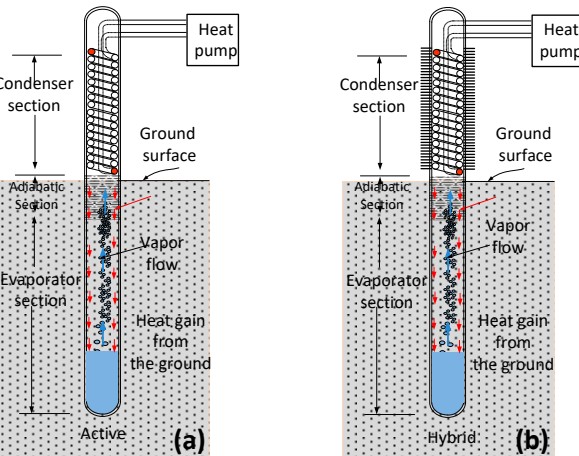

**Figure 3.** Schematic of (**a**) active and (**b**) hybrid systems.

### 1.2. Main Stages of GCT Development

In general, the history of GCT studies is characterized by significant inconsistencies and differences in both nomenclature and underlying theories or methodologies. These incompatibilities stem in part from the diverse applications and backgrounds of the researchers and the practitioners (e.g., mechanical, civil, soil scientists, hydrologists, and hydrogeologists) in this field. However, it is not the objective of this section to detail the history of the thermosiphon technology, but rather to provide an overview of the most important stages through which this technology has evolved. The invention of the two-phase thermosiphon dates back to the late 1800's with the patent of the Perkin's tube, a basic form of the thermosiphon. Overall, the first improvement of the original ideas was patented by Gauglerinin 1942 [18]; however, it was only after the early 1960s that the thermosiphon took practical shape. By this time, its remarkable properties were fully appreciated and serious development work took place [19]. Since then, the thermosiphon technology has been adapted to be used in geothermal activities and significantly improved, going from a vertical heat transfer device (1960) to sloped (1978) and flat loop designs (1994). Nowadays, even completely buried designs are proposed (1998). The first devices used were vertical sealed tubes installed into the ground with the condenser at the surface. A further advancement was the hairpin system (1998), in which the thermosiphon is entirely buried beneath the ground surface. The thermopile is presently commonly used to support structures on piles within frozen ground. An application example is the project of the Alaska pipeline, built around 1975 [20]. All of these improvements have resulted in very efficient devices and have contributed considerably to solving several engineering construction, resource exploitation, and environmental protection issues,

especially in cold regions. A chronologic flowchart of the thermosiphon technology development is presented in Figure 4. Figure 5 shows a few of the thermosiphon types reported in the literature. A review of different cold climate systems was first presented in the paper of [11] and recently in that of [8]. The review included the literature from the previously published work on the different structures and arrangements of GCTs. Table 1 summarizes several configurations of GCTs provided in the literature. This table includes the type of technology, main application/characteristics, and appropriate references. As already indicated, a further advancement of the GCT was the Geothermal Convector (GC), proposed by Carotenuto et al. [13] (Figure 6a), which was developed for geothermal power production in the early 1990s [3]. More details regarding this technology are given in Table 1.

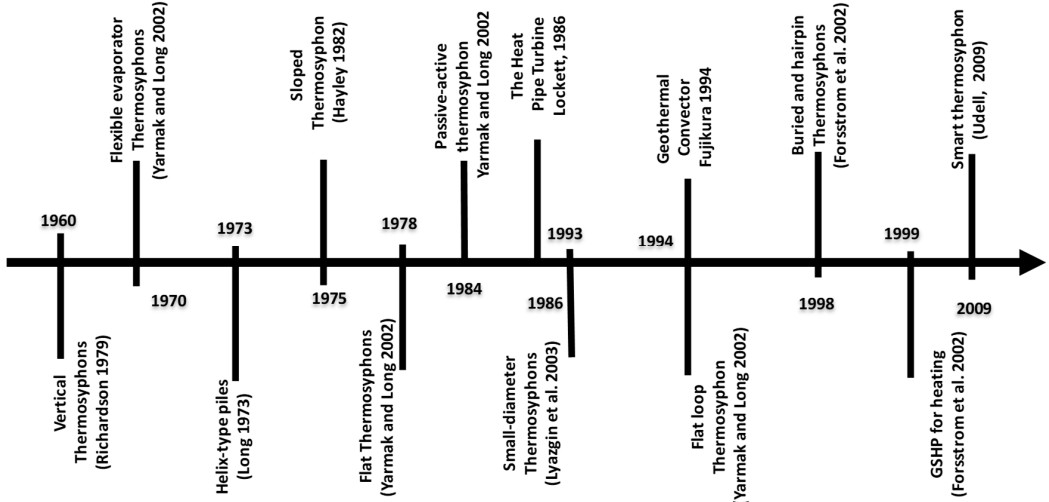

**Figure 4.** Chronologic flow chart of the thermosiphon technology development.

An interesting type of GCT technology is the Heat Pipe Turbine (HPT), thermosiphon Rankine engine [21], or Thermosiphon Rankine Cycle (TRC) [22], proposed by Lockett [23]. The HPT is based on the same principle as the typical Rankine cycle, which is used to generate power with temperatures in the range 80 to 150 °C. A summary of various HPT systems found in the literature was given by [3]. The basic configuration includes a closed vertical cylinder working according to the thermosiphon principle. The turbine is placed close to the upper end, between the adiabatic section and the condenser section. A plate is used to separate the higher pressure region from the lower pressure region. Conversion of the fluid enthalpy to kinetic energy is achieved through a nozzle. Figure 6b shows a schematic view of the HPT. An improved TRC system was developed by [22], where vapor and liquid flow passages are separated by installing a liquid feeding tube with showering nozzles. In short, HPT and TRC systems have largely remained at the concept stage; they have not reached the stage of maturity. The current few studies are not yet sufficient to obtain a satisfactory scientific appreciation of the technology.

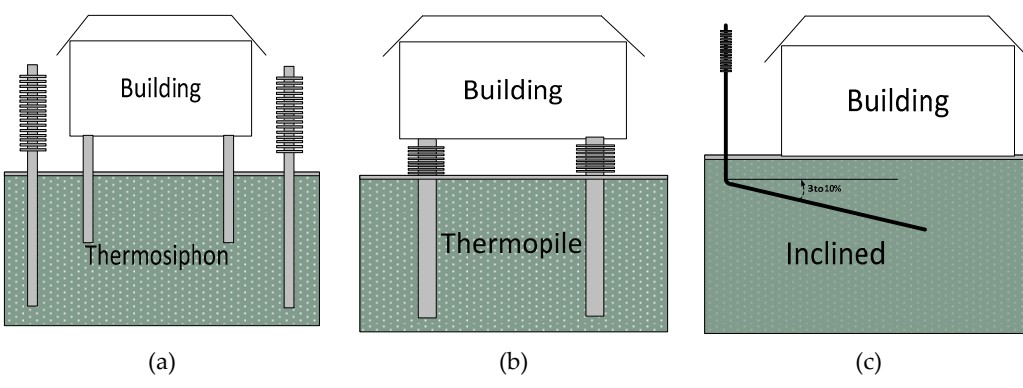

(a)          (b)          (c)

**Figure 5.** *Cont.*

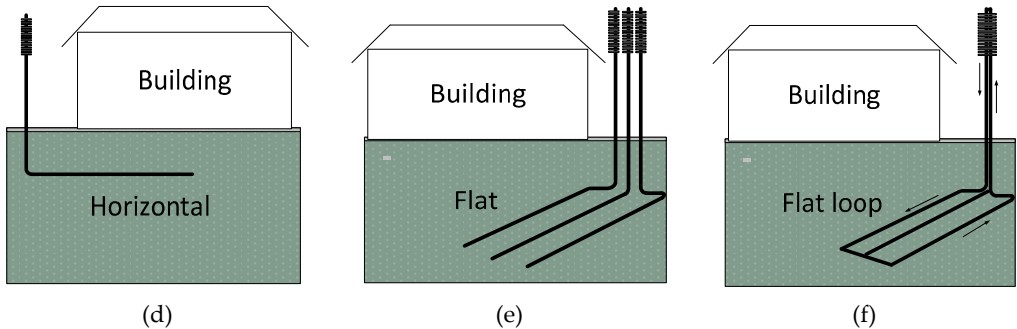

**Figure 5.** Schematic of major GCT designs modified from Holubec [24] and Xu and Goering [11].

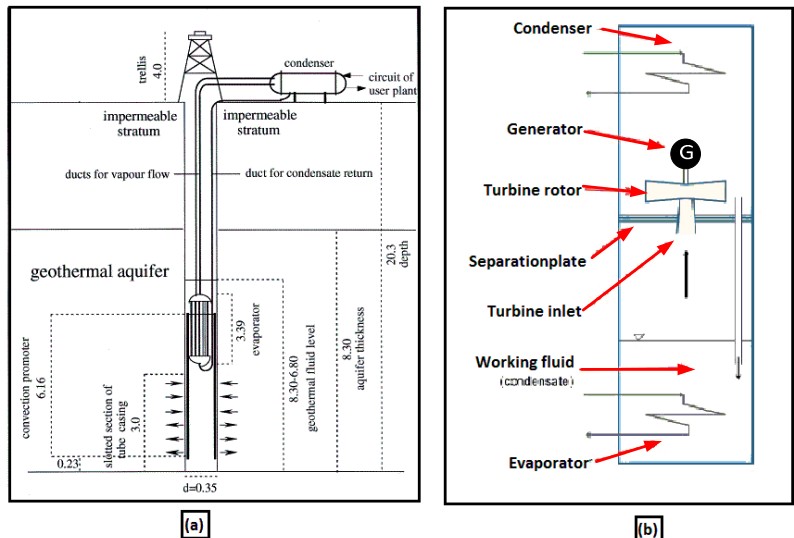

**Figure 6.** (**a**) Geothermal convector (GC) from [13] and (**b**) heat pipe turbine from [3].

### 1.3. Latest Developments

This section presents important advancements in GCT technology. A summary of these latest developments is presented in Table 2.

### 1.3.1. GCT-Assisted Heat Pump (GCT-HP)

This category is an important advancement of GCT technology. Originally, GCT-HP units were designed as hybrid or active systems, so that they could speed-up the ground freezing for different applications. Later, in the beginning of 1998, they were proposed for heat extraction from the ground for domestic space heating [1]. In 1998, a patent was filed for the use of $CO_2$ in vertical earth probes [25]. Since then, several improvements have been made on this prototype [26]. The advantage of a thermosiphon using carbon dioxide as the working fluid in conjunction with a Heat Pump (HP) was shown by Ochsner [27]. During the winter of 2006–2007, the GCT-HP was able to successfully heat a home residence with an average Coefficient of Performance (COP) of 4.1. A research and development status on $CO_2$ earth heat pipes for GSHPs enhancement was presented by Kruse and Russmann [28]. Both vertically and horizontally, ground thermosiphons in conjunction with HPs are studied. In order to guarantee a small pressure drop and good heat transfer at the operating conditions, different investigations were carried out on condenser design ''probe heads''. For the evaporator, serpentine and spiral geometries were proposed by Rieberer [29]. Acoaxial design, where the liquid phase flows down through a central pipe and the vapor flows upwards through an annular channel, was suggested by Oschner [27]. A corrugated stainless steel heat pipe was introduced by Kruse and Russmann [28] and a U-pipe loop by [1,30]. Current GCT-HP evaporators have a length of 100 to 150 m, which is sufficient to supply an HP for a single-family house. To meet the higher thermal

power needs of urban heating areas, [31,32] successfully realized a 400 m deep $CO_2$ thermosiphon installation. Regarding the condenser, a coaxial design was initially used, but it was very sensitive to capacity variations. Later, a collector head was developed with a plate heat exchanger and an external liquid separator. In addition, a special $CO_2$ condenser was developed (Figure 7), to better connect the thermosiphon to the heat pump. This special $CO_2$ condenser design consisted of a shell and coil set-up with two parallel heat exchangers inside a pressure vessel [28], ensuring a smooth film at the inner tube wall condenser. A transverse cross sectional view in the condenser of the passive-active thermosiphon patented by Haynes et al. [33] is presented in Figure 8. In light of these developments, the GCT-HP appears to offer a very interesting application potential not only for single-family houses, but also for high capacity urban heating systems.

### 1.3.2. Smart or Reversible GCT

This is an innovative concept of thermosiphon technology used to transfer heat to and from the soil [34]. The thermosiphon is supplemented by a liquid pump (Figure 9). In the heating mode of operation, the working fluid extracts heat from the soil and releases it at the device's top when an above ground heat exchanger (exposed to cooler ambient air, room air, or connected to a heat pump) is used. In this mode, soil temperature has to be higher than that of the heat exchanger temperature (i.e., the undisturbed ground temperature). However, in the cooling mode, the pump displaces the cold liquid refrigerant from the bottom of the thermosiphon and transfers it to the heat exchanger. In this case, the liquid evaporates in the heat exchanger at a higher temperature, producing a cooling effect. The vapor then returns back to the colder bottom part of the thermosiphon to condense, releasing heat to the soil. The advantage of a reversible thermosiphon was shown by Kekelia and Udell [35]. This technology was suggested by Jankovich [36] as a thermal energy storage system. Overall, the concept of reversible GCT was shown to be promising in both heating and cooling modes; however, it still remains in the early stages of development.

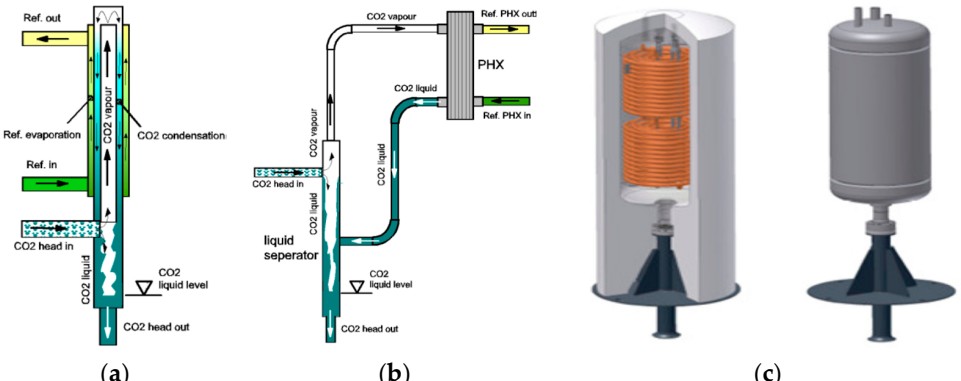

**Figure 7.** $CO_2$ condenser type: (**a**) the coaxial type [37], (**b**) the plate heat exchanger type [37], and (**c**) the $CO_2$ shell and coil condenser type [28].

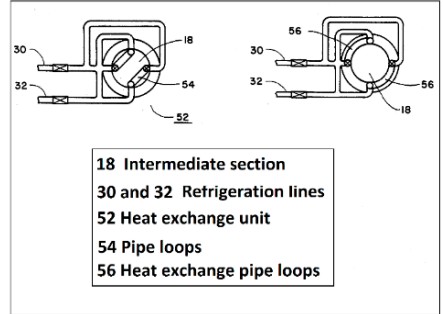

**Figure 8.** A transverse cross sectional view in the condenser of a passive-active thermosiphon from [33].

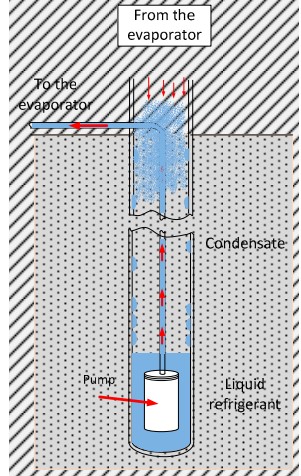

**Figure 9.** Smart or reversible thermosiphon modified from [36].

### 1.3.3. Polymer Thermosiphons

Another relatively new tendency in thermosiphon technology is related to nano composites, which have opened up new opportunities for GCT development, design, and use. This technology was developed to produce loop polymer thermosiphons capable of long-term operation. The evaporator and the condenser of such a thermosiphon have the form of slabs located horizontally in the soil at different depths. They are connected by flexible polymer pipes used for the transmission of vapor and liquid streams [38]. Plastics offer increased corrosion resistance and an increase in flexibility. The plastic evaporator may be used in either ConventionalSloping Evaporator (CSE) or Flat Loop Evaporator (FLE) systems, depending on the outcome of the prototype evaluation [11]. It was found that the evaporator thermal resistance of polymer thermosiphon is similar to that of a classical aluminum heat pipe [39]. Across section of a Super-Long Heat Pipe (SFHP) made of thermally conductive polymer, lined with a metallic membrane and thermally conductive glue, anda flat loop thermosiphon made from a polymer composite, are presented in Figures 10 and 11, respectively.

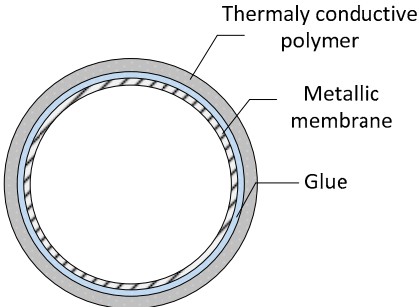

**Figure 10.** Cross section of SFHP made of a thermally conductive polymer, lined with a metallic membrane and thermally conductive glue [40].

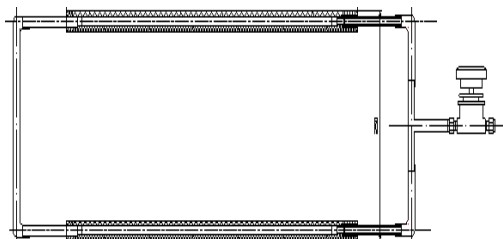

**Figure 11.** Flat loop thermosiphon made from a polymer composite [41].

### 1.3.4. VapordynamicThermosiphons

An interesting type of thermosiphon technology is the Vapordynamic thermosiphon (VDT). This technology, patented in 1985 [42] has been suggested for heating floors in houses, timber drying, roof snow melting, or as a thermal control system for snow thawing. The VDT differs from the conventional thermosiphon of the same diameter in two ways. The first is related to its horizontally oriented annular condenser and the two-phase flow structure of the working fluid inside it [41], which makes it almost isothermal with the length of tens of meters [43]. The latter is related to its spatially separated vapor flow and two-phase liquid flow (tube in tube heat exchanger), which avoids a negative interaction between the opposite flows of the vapor and liquid encountered in conventional thermosiphons. A complete review of VDT designs and applications, including the schematic, principles, main parameters, and thermal control of different heat loaded devices, is given in [41,44].

### 1.3.5. Thermosiphon with Internal Screw Insert

As indicated by Vasiliev et al. [44], a thermosiphon with an internal screw was proposed in 1985 [45]. A helical insert was placed into the thermosiphon to create a swirl in the vapor and liquid flows. This produces artificial turbulence in the vapor flow and hence, intensifies heat transfer in the condenser and the evaporator. A major disadvantage of the helical insert is the somewhat increased pressure drop in the vapor flow. The helical insert may be built of a different material, such as metal (aluminum, steel), porous ceramic, or a polymer composite. More details on this thermosiphon with an internal screw can be found in [44].

### 1.3.6. Thermosiphon Heat Transfer Device with Bubble Driven Rotor

An interesting development in thermosiphon technology is that of a device integrating a bubble driven rotor patented by Read et al. [46]. The rotor may be of any suitable shape and dimensions, including conveyors rotating around two or more axes. Buoyancy forces, bubble formation, and growth serve as the main energy movers driving the rotor. In accordance with the application objective, this set-up may be optimized for transferring heat, generating power, or a combination of both.

**Table 1.** Summary of several configurations of natural convection GCHE.

| Type | | Application/Characteristics | Principle/Details |
|---|---|---|---|
| **Vertical** | Thermosiphon | • Diverse applications, but mainly used to stabilize constructions in arctic climates.<br>• Do not carry any load | • Transfer 40W/m minimum.<br>• Diameters of the heat pipes used are 3 and 7.5 cm and lengths vary between few meters to 400 m [31]. |
| | Thermopile | • Support structures on piles within frozen ground<br>• Load bearing capacity [47] | • Generally made of 4 to 12NPS steel pipe [48].<br>• Initially, propane was the refrigerant of choice, but it was replaced by $CO_2$.<br>• Undergoes several changes from the basic design [8]. |
| | Double coaxial tube | • Can be coupled with $CO_2$-HP for domestic space heating [27]. | • Has not been tested yet.<br>• A separation of the $CO_2$ liquid and gaseous phases takes place.<br>• The vapor phase of $CO_2$ ascends in the outer pipe to the heat exchanger (evaporator of the heat pump), condenses and runs down in the inner pipe [27]. |
| | Helix and U-pipe loop | • Heating applications [1,49]. | • Only 22 mm insulated pipe brazed together at the bottom with a 28 mm riser tube that has been tested [30]. |
| | Closed loop | • Large scale extraction of geothermal energy [3].<br>• Power production. | • 150 mm outer diameter, and 150 m length [50]<br>• Generate power with temperatures in the range 80 °C to 150 °C. |
| **Sloped ST** | | • Employed in the majority of passive subgrade cooling system installations beneath slab-on-grade [51]. | • Pipe is normally a 100 mm OD.<br>• Typically, evaporator slopes are between 10% and 3% with the median being 5%.<br>• Low suspected of being damaged by excavation. |
| **Flat FT** | | • Prevent thaw settlement problems example roadway embankment [24]. | • The grade of the evaporator needs to be absolutely horizontal to ensure a uniform distribution of the liquid phase of the working fluid [11]. |
| **Flat loop** | Slab-on-grade | • Used for slab grade-on design or a crawl space installation [24].<br>• Used to keep the foundation frozen at the bottom of dams [8]. | • Common length 178 m and 150 m [24].<br>• Pipe is normally a 50 mm OD, loop lengths long as 600 m has been installed.<br>• Typically theses small evaporator diameter, made of 3/4" steel pipe weighs less than similarly sized sloped thermosiphons.<br>• Small diameter of the tubing, making it easier to damage during excavation.<br>• Typically, the loop level is 0.5 m to 1 m below the base of the subgrade insulation. |
| | Crawl-space | | |
| **Flexible evaporator** | | • Used for ice and snow melting in Japan and Turkey [52].<br>• Proposed in hybrid system using HP and a super-long flexible heat pipes [53]. | • The materials that have been used for the flexible sections include annealed copper, aluminum extrusions, and coiled steel oilfield tubing.<br>• Capability of being fully charged in the shop and coiled or folded for shipment and then formed into their appropriate configuration for the installation at the project site [11]. |
| **Buried and hairpin** | | • Used generally for road and runway installations [54,55] e.g., Thompson Drive project. | • Entirely buried beneath the ground surface.<br>• The condensers can be coupled to either vertical or sloping evaporators.<br>• Overall cost is less than that of conventional systems [54].<br>• This type of configuration eliminates the safety and esthetic issues [54]. |

**Table 2.** Summary of the latest development in GCTs.

| Type | Main Application/Principle | Finding/ Characteristics | Refrigerant |
|---|---|---|---|
| **GCT-HP** | • Used for heating new single-family houses in Austria [29]. | • Maximum heat extraction rate is 50 W/m | • R22, R134a and R290 (propane). Finally, $CO_2$ is adopted. |
| **Smart thermosiphon** | • Used for heating and cooling mode applications.<br>• Seasonal Underground thermal Energy storage [36,55]. | • In these systems returns liquid condensate to the heat exchanger at a rate determined by the mass flow rate of vapor entering the thermosiphon. | • R134a |
| **Vapordynamic thermosiphons** | • Used for the floor heating in houses, timber drying, roof snow and control systems for snow thawing melting [43]. | • Low sensibility to the working fluid quantity, presence of a noncondensing gas - Possibility of various embodiments with an extensive zone of evaporation including bent or flexible modifications.<br>• The VDT with porous coating ensures a heat transfer enhancement up to 5 times compared to the plain tube thermosiphon [44]. | • Ammonia, R600, water and propane [44]. |
| **heat pipe turbine** | • Only small size power generation units (50 to 100 kW) [3,22]<br>• Only some design schemes or prototypes are available. | • Overall efficiency is less than 3%.<br>• Capable of operating at low-grade geothermal heat sources with temperatures as 55 °C but with only 0.1% overall efficiency.<br>• Reducing vibration will improve the efficiency performance [56]. | • Water, Isopentane, R123, R114 [3]. |
| **Polymer thermosiphons** | • Recommended for heating the floors of storehouses for vegetables and other agricultural products, heat exchangers contacting with ground heat pumps [57].<br>• Proposed for snow melting [40,52]<br>• Protection of the thermosiphons from corrosion | • The distance between the evaporator and condenser can constitute several meters.<br>• Long service life compared to metallic ones.<br>• Can be completely buried devices. | • Isobutene (R600) [58].<br>• R123, R134a, and ammonia [52]. |
| **Screw thermosiphons** | • Enhanced heat transfer in the condenser and evaporator [44] | • Decrease the thermal resistance of the thermosiphon down to 30–50%. | • Acetone [44]. |

## 2. Thermosiphon Applications

GCT applications are numerous and varied. A review of the literature reveals that the application of thermosiphons is so extensive that it is not possible to give a complete review and evaluation in this paper. The first and still most prevalent application is to provide thermal stability and support to foundations constructed on permafrost in cold regions. Most of these applications have been deployed in arctic climates [8]. They have also been used for the stabilization of road and rail embankments and to create a frozen barrier for the containment of contaminants [16,17]. Further applications are found in roadways and bridge heating to remove snow and ice, cooling soil in permafrost locations to improve its mechanical strength [59–62]. Fukuda et al. [63] and Mashikoet et al. [64] suggested the use of thermosiphons in cold storage systems for agriculture product preservation. Applied hybrid systems (GCT-HP) solutions for house-heating purposes are adopted in Germany and Austria [27,37,65,66]. A smart heating and cooling system based on the thermosiphon principle was used by [34–36,67,68]. GCT technology was also successfully implemented in mega-projects, such as the Alaska pipeline, where over 120,000 thermosiphons were installed for an oil pipeline crossing Alaska. The Qinghai-Tibet Highway (QTH) relies on thermosiphons to mitigate thaw subsidence of foundation soils [69], as well as to maintain and upgrade embankments along the QTH. GCTs have also been successfully employed in the 632 km-long Qinghai-Tibet Railway (QTR) [70] since January 2011, and in the relatively recent-built China–Russia Crude Oil Pipeline (CRCOP) [71]. Another project is the Power Transmission Line (TPTL) along the Qinghai–Tibet Power Transmission Line, completed in 2011 [72]. This project has a total of 2361 transmission towers, 1207 (51%) of which are located in permafrost regions. The de-icing and snow melting system, based on the thermosiphon principle, is a concrete example of the GCTs application [73]. Most recent thermosiphon applications have been in nuclear waste sites [74], mining, tunnels [8,75], and containment barriers for hazardous waste and dams. They have even been considered in more unconventional applications, such as preserving archeological sites in the Arctic.

Based on the available literature, a classification of the ground coupled thermosiphon applications is proposed in Figure 12. From this figure, two major applications for the exploitation of medium and low temperature geothermal sources can be identified. Medium temperature applications are generally used for the large-scale extraction of geothermal energy for direct useand power production. This case was critically reviewed and analyzed in 2013 by Franco and Vaccaro [3], revealing that only a few design schemes and prototypes are available in the literature. To the best knowledge of the authors, since that time, no work has been done on a large-scale on geothermal energy extraction for direct use and power production by means of GCTs. Regarding low temperature applications, GCTs are mainly used for temperature control and heating and cooling applications. It may be observed that improvements to and new developments in low temperature GCT applications are still the subject of ongoing research. Current applications are often designed to meet environmental demands and the stability requirements of engineering infrastructures in cold regions. However, under the global warming scenario, the "hybrid GCT" may represent the best design approach for important constructions in a permafrost environment. Such a concept changes the design method from passive to proactive cooling in order to better deal with the impact of global warming on permafrost.

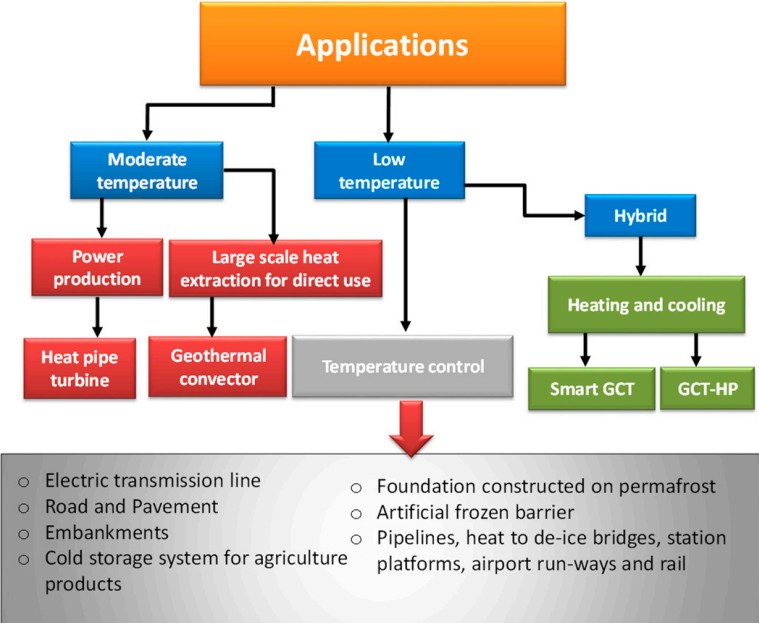

**Figure 12.** Classification of thermosiphon applications.

## 3. Performance of GCTs

The GCT thermal performance assessment is very complex because it is a device where several parameters dominate. The thermal performance of these systems depends on various factors summarized below:

1. **Climatic conditions**: ambient temperature, solar radiation, wind effect, rain, humidity, sky temperature, coolant flow rate and temperature, site constraints such as orientation, tilt, and surroundings;
2. **Thermosiphon characteristics:** material properties, surface coatings, absorptance, geometry such as diameter and length of various sections (evaporator section, adiabatic, and condenser sections), working fluid properties, filling ratio, and internal working pressure;
3. **Ground thermal properties**: temperature, thermal conductivity, and thermal diffusivity;
4. **Other conditions**: load characteristics related to the application and climate warming.

The thermal performance of GCTs can be distinguished by the heat transfer rate per system unit length and time, the COP (in some case of hybrid systems), or the working period in some other cases. In this paper, the thermal performances of GCTs are organized in two separate subsections: (1) Mathematical modelling, simulations, and parametric studies; and (2) experimental studies.

### 3.1. Mathematical Modeling, Simulations, and Parametric Studies

#### 3.1.1. Modeling and Simulation

Given the wide range of applications of GCTs, demands for modeling are diverse. In order to respond to all the requirements, a variety of analytical and numerical models and simulation tools have been developed to assist in the design of GCTs. Some researchers have presented different modeling tools, ranging from simple design and analysis to complex transient 2D and 3D simulations. One common approach, referred to in several papers as the coupled air-two-phase closed thermosiphon-soil heat transfer model, relies on a coupled heat transfer model (air-GCT-soil). Table 3 illustrates the main GCT mathematical modeling, simulation, and parametric studies available in the literature: they are presented in chronological order in terms of their main features and specifications, special findings, and related comments.

Modeling the temperature in the surrounding soil zone is an area where significant effort has been made. At present, two different approaches are found in the published literature. The first one considers the ground as a semi-infinite medium, with constant properties and heat transfer outside the borehole occurring only by conduction. Most models are based on either analytical or numerical methods. A few models are based on the combination of the analytical and numerical solutions, such as Eskilson's model [76]. A complete review of various analytical and numerical models has recently been published by several authors [4,5,77–79]. Briefly, this modeling approach has undergone many improvements in recent years, starting from the infinite line source model (ILS) [80], the infinite cylinder source model (ICS) by Ingersoll et al. [81], the finite line source (FLS) model by [82] and [83], to more complex two- and three-dimensional finite volume or finite element numerical models [84–86].

Any of these models have to be first validated prior to generating reliable simulation results. Two validation approaches have mainly been used in the literature: (1) use of experimental measurements with an allowable accuracy to compare withmodel predictions under given conditions and (2) validation by inter-model comparison of analytical and numerical solutions.

Short-term response models are often validated using data from a small- or medium-scale laboratory setup (Sandbox) and in situ thermal-response tests (TRTs). Such models correspond to time periods ranging from a few minutes to a number of days.

Data for long-term response models are rare. Advantages of indoor sandbox experiments include well-controlled parameters, which enable a complete and thorough validation of the model. Unfortunately, very few indoor sandbox experiments have been reported [87,88]. The lack of accurate values of the ground thermal properties is another serious obstacle in model validation because of various uncontrollable testing uncertainties and estimated equivalent properties that are generally obtained by curve-fitting approaches and some model simplifying assumptions. For instance, ground thermal conductivitiesare determined by TRTs with an accuracy of about 10%.

**Table 3.** Summary of previous analytical and numerical studies of GCTs.

| Publications/Refs. | Objectives/Purpose | Method Used | Conclusions/Special Findings |
|---|---|---|---|
| [89] | Performance simulation of thermosiphon to mitigate thaw settlement of embankment in sandy permafrost zone | • 3D Air-thermosiphon-soil coupled model.<br>• Use a TRM for the thermosiphon | • Thermosiphon provides rapidly a cooling effect and approaches a thermal balance state after 10 years of construction |
| [53] | Proposed hybrid GSHP- SFHP system | • Use 2D CFD (ANSYS) for ground temperature in modeling and TRM for the thermosiphon | • $L_e$ is the most important aspect that influences the heating performance of the SFHP and the temperature recovery in ground. |
| [90] | Calculations for thermal stabilization of transport embankments and their bases | • Calculations carried out for two years of operation.<br>• Only half of the section was studied.<br>• Use of inclined thermosiphons | • Calculations of change of temperature fields in the soil are insufficient when designing the thermal stabilization of soil. Forecast of the deformations of heaving and thawing is necessary. |
| [40] | Simulation of the heat transfer in pavement with polymer super-long flexible heat pipes (SFHPs) melting snow. | • A 2D model is developed for the pavement<br>• Use TRM for the SFHP | • Ammonia is optimal for SFHPs concerning the maintenance limit.<br>• Recommended dimensions of an SFHP are ø 32 × 2mm. Le = 40 m, Lc = 20 m. |
| [91] | Study of six different forms of pile arrangement effect on The Temperature Control Technology of bridge foundation in permafrost regions for the next 50 years. | • 3D numerical analysis was used by finite element software ANSYS | • The same number arrangement of the thermal pile is better outside the pile foundation than that inside the pile foundation. |
| [92] | Numerical simulation of the thermal conditions of two phase codes thermosiphon embankments affected by the shady-sunny slope effect. | • Use coupled air-TPCT-soil heat transfer model.<br>• 3 D control volume integration method is used. TRM. | • both the unilateral and bilateral can cool the permafrost stratum, but the unilateral aggravates the asymmetric geo-temperature caused by the shady-sunny slope effect<br>• A permafrost embankment with bilateral TPCTs is able to reduce the shady-sunny slope effect. |
| [31] | Simulation of a 400 m vertical $CO_2$ heat pipe for geothermal application | • A comparison of the helically corrugated pipe and plain pipe. | • Simulations were performed only for the plain pipe thermosiphon<br>• Simulation for steady state conditions reproduced the heat transfer and situation of flow within the plain pipe thermosiphon.<br>• The heat transfer in the pool region is lower than in the film region, which was confirmed by the measurements. |
| [32] | Dynamic Simulation and validation of a 400 m vertical $CO_2$ heat pipe for geothermal application | • A quasi-dynamic model including mass transfer between the liquid and vapor phases as well as the conduction heat transfer from the surrounding soil towards the pipe. | • Model validation against experimental data. |
| [93] | Application of heat pipes on geothermal heat pump system (HP-GSHP).The characteristics of vertical closed-loop ground source heat pumps were compared in a typical type, direct expansion type, and heat pipe type, respectively | • Developed a simple mathematical model to evaluate the annual performances of the three different types of ground source heat pumps for Anchorage, Ottawa and Seoul. | • HP-GSHP Reduce energy consumption for heating in Seoul (Korea) by 10.3% than DX-GSHP and 21.1% than SL-GSHP. |

**Table 3.** *Cont.*

| Publications/Refs. | Objectives/Purpose | Method Used | Conclusions/Special Findings |
|---|---|---|---|
| [94] | Modelling the crack formation of a highway embankment installed with two-phase closed thermosiphons in permafrost regions | • Developed 2-D and 3-D finite element models with the assistance of COMSOL. | • Suggestions are proposed to reduce the problem of the crack formation of a highway embankment. |
| [95] | Control the ground temperature for a tunnel section in a permafrost region. | • A composite heat transfer model, including the air inside the tunnel, the air-TPCT-soil group | • GCTs can cool down the soil layers in the shallow tunnel section and ensure the thermal stability of the tunnel. |
| [96] | Numerical analysis on the thermal regimes of thermosiphon embankment in snowy permafrost area. | • 3 D Finite element method (FEM).<br>• Temperature boundary is determined according to field observation data in Mohe, China. | • GCT cannot avoid the degradation of permafrost under snow cover.<br>• Composite measures need to be adopted to keep embankment stability in snowy permafrost zones. |
| [69] | To study the long-term cooling effects of thermosiphons around tower footings along the QTPTL | • Improved 3-D numerical coupled heat transfer model.<br>• 50-year operational period simulations. | • During a 50-year operational period, foundation soils under the footing would roughly go through a rapid cooling stage in the first to 5th year, a stable stage in the 6th–15th year, a rapid warming stage in the 16th–35th year, a slow warming stage after the 35th year. |
| [97] | Prevention of icing with ground source heat pipe | • A theoretical analysis for Turkey's climatic conditions | • Increase of heat pipe diameter decreases the Re number. This the heat flux increases.<br>• Increase of ground temperature increases the Re number the flow increases thermal resistance decrease. |
| [98] | Simulation of the thermal performance of a combined cooling method of thermosiphons and insulation boards for tower foundation soils along the QTPTL | • 3-D numerical coupled heat transfer model for the air–thermosiphon–soil system. | • To retard thaw penetration around the footing of tower foundation, a combined cooling method of thermosiphons and insulation boards is proposed. |
| [99,100] | Examination of the influence of outer thermal resistances in the pipe, borehole filling, and surrounding sub-surface on the performance of a partially wetted geothermal thermosiphon. | • Parametric investigation uses a quasi-three-dimensional numerical model. | • Pipe and grout material becomes increasingly important for lower wetting ratios and shorter extraction times.<br>• A well-wetted polyamide plastic pipe with a thermal conductivity larger than 2 W/(m K) would be a possible alternative to steel pipes. |
| [101] | Feasibility study for using thermosiphons with pipelines in arctic regions to reduce the potential for frost heave. | • 3D- FEM<br>• Study proposes an anisotropic conduction model that simplifies the thermal-fluid processes within the thermosiphonwithout overwhelming computational cost. | • Introduced a method that can be used to optimize the design of new infrastructure and pipelines in permafrost, as well as to assess how thermosiphons mitigate frost heave in existing projects. |
| [102] | Numerical simulation of heat transfer processes in cone-cylinder pipe and cooling effects of thermosiphon along the Qinghai-Tibet DC Interconnection Project | • 3D-FEA Coupled heat transfer model among air, thermosiphon and soil is developed. | • Quickly cooling processes of ground temperature near the pipe occur mainly within the first 5 years after application of the thermosiphon. |

**Table 3.** *Cont.*

| Publications/Refs. | Objectives/Purpose | Method Used | Conclusions/Special Findings |
|---|---|---|---|
| [103] | Applications and analysis of a two-phase closed thermosiphon for improving the fluid temperature distribution in wellbores. | • Parametric study, in which a number of influencing have been investigated on the efficiency of the thermosiphon. | • Demonstrated that the TPCT techniques can be feasible and effective means of enhancing oil production rates. |
| [104] | Investigation on the feasibility of periodic two phase thermosiphons for environmentally friendly ground source cooling applications | • A semi-analytical approach is used to simulate the transient behaviour of the system. | • Device may be promising for ground source cooling applications. |
| [105] | Investigation of partially wetted geothermal heat pipe performance | • Parametric study, using transient, 2D numerical model used.<br>• Effect of different parameters on the performance of partially wetted geothermal heat pipes is studied. | • Borehole diameter, borehole filling thermal conductivity have a strong influence on polyamide heat pipe performance.<br>• Soil thermal conductivity is of minor importance for short times. |
| [106] | Analysisof heat transfer in thermosiphons and U-tube ground source heat pumps | • A simple analytical heat transfer model<br>• heat transfer from a thermosiphon is considered under steady state | • Heat transfer rate per unit length associated with the thermosiphon was found to be approximately 2.3 times greater than that of the U-tube system. |
| [107] | Investigation of the cooling effect of combined L-shaped thermosiphon, crushed-rock revetment and insulation for high-grade highways in permafrost regions | • Combined an L-shaped thermosiphon and insulation into a crushed-rock revetment. | • The combination of the L-shaped thermosiphon, crushed-rock revetment and insulation is effective to keep the thermal stability of the large-width embankment in its service life under the condition that the air temperature is warmed up by 2.6 °C. |
| [108] | Numerical investigation of the cooling characteristics of two-phase closed thermosiphon embankment in permafrost regions | • Use a 3D Air-thermosiphon-soil coupled model.<br>• Use a TRM for the thermosiphon.<br>• Use the method of sensible heat capacity for the soil. | • Thermosiphonswere recommended to be used in new and/or in the maintenance of the existing constructions in permafrost regions. If combined with other engineering methods, e.g., insulation, crushed rock, etc., thermosiphons will present a more effective devices to ensure the stability of constructions in warm permafrost regions. |
| [109] | Power generation capacity study of an EGS configuration using a thermosiphon. | • A coupled finite-difference wellbore and reservoir model.<br>• The effects of several factors have been examined. | • Insulation of the inner tubing is critical to transporting sufficient heat to the surface.<br>• Without the benefit of fracture circulation the system would be of modest power generating capacity and would decline quickly. |
| [110,111] | Thermal performance study of heat pipe arrays In soil. Evaluated the effect of spacing, type of array, heat pipe properties, and soil properties. | • Transient mathematical model<br>• Parametric study | • Soil thermal conductivity and spacing of heat pipes have significant effects on the temperature rise of the heat pipes. |
| [101] | Analysis of forced-air and thermosiphon cooling systems for the Inuvik airport expansion | • Use a two-dimensional thermal program, THERM2 | • Based partially on the results of the analyses, it was decided to utilize air ventilation for cooling. |

Overall, validation with experimental measurements or by inter-model comparison is a common scientific approach in the process of model development. It has been addressed at various degrees of detail by many researchers, typically [112,113], to name only a few. The literature reveals that numerical models can offer a higher degree of accuracy (especially on short-term scales) than analytical models, due to the assumptions which neglect some of the underlying physical mechanisms. However, the required computation time of analytical models is much shorter, compared with numerical models. For example, Saqib and Claesson [114] compared new analytical and numerical solutions. Such comparisons showed that the agreement accuracy between bothsolutionswas higher than 0.01 K. Furthermore, some models are only valid after a recommended minimum time. It was reported that using the Kelvin's line source may cause a noticeable error when $\alpha t/r^2_b < 20$ [79]. Comparison of a composite-medium model with a set of reference sandbox data demonstrated that this model was suitable for times as short as several minutes [115]. Min and Lai [77] reported a comparison of six analytical models for a time interval that ranged from several minutes to decades: an infinite cylinder-source model, two infinite line-source models, two finite line-source models, and a composite-medium line-source model. The conclusion was that all the models examined could be used to predict the medium-term temperature responses. However, the composite-medium line-source model should be used to predict the short-term responses, and the finite line-source models should be used to calculate the long-term responses.

The second approach considers the ground as a water-saturated porous medium involving melting/solidification phenomena. This approach is the most used in GCTs modeling, compared to the semi-infinite, constant properties concept for the ground. Heat transfer problems including the moving boundaries of the melting/solidification are known as Stefan's problem [36,80]. The literature dealing with the freezing of a liquid saturated porous medium is abundant and an extensive review is available [116]. Generally, the analytical modeling of such problems requires many simplifying assumptions. The models are usually one-dimensional, limited to conduction heat transfer, use simple initial and boundary conditions, and assume constant thermophysical properties (e.g., density) [117]. Since the treated problem is nonlinear, it is generally solved analytically only for a limited number of cases [80]. For this reason, the majority of the available research resorts to numerical methods, using numerical schemes with finely divided meshes around the GCT, especially those regions near the tube. A summary of analytical and numerical studies of GCTs reported in the literature is illustrated in Table 4. The physical model of the soil heat transfer process around the GCT is presented in Figure 13 [112,118].

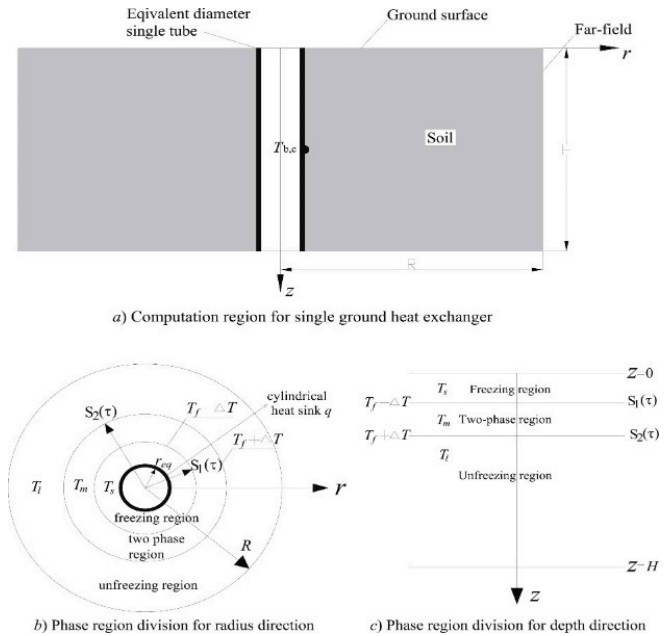

**Figure 13.** Physical model of soil heat transfer process around the GCT [112].

In the processes of melting and solidification, the general governing equations consist of energy balance equations. Taking phase change into consideration, and ignoring water infiltration into the soil and water motion in the active layer, the energy balance equation is reduced to one differential heat conduction equation (Equation (1)). The basic form of this governing equation has been employed in Cartesian 3D [55] and in cylindrical coordinates [72] for the solid and liquid regions. In most cases, the simplest technique used to consider phase change through the heat conduction equation is the so-called ''effective-heat capacity'' method, in which an equivalent sensible heat capacity, as defined by Equation (2), is substituted for the latent heat [119]. An alternative modeling technique can be established on the basis of the enthalpy that is included in the energy equation, as shown in Equation (3) [36]. Finite difference (FD) or finite element modeling (FEM) techniques have been used [115] to model the heat flow around the CGT. The Galerkin method is commonly applied to obtain numerical solutions, as explained in [55,72] and [96,120]. However, for the control-volume integration technique, the discretized equations are solved in an iterative manner using the Successive Under-Relaxation Method [108].

$$\rho C p \frac{\partial T}{\partial t} = \nabla.(k \nabla T) \tag{1}$$

$$C_p^{eff}(T) \begin{cases} Cp & T < T_m - \Delta T \\ \frac{L}{2\Delta T} + Cp & \text{at } T_m - \Delta T < T_m + \Delta T \\ Cp & T > T_m + \Delta T \end{cases} \tag{2}$$

$$\frac{\partial H}{\partial t} = \nabla.(k \nabla T)$$

Where $H$ represents the enthalpy per unit mass, which is defined by the following relation:

$$\tag{3}$$

$$H = \begin{cases} C_{ps}T & T < T_m \\ C_{pl}T + \left(C_{ps} - C_{pl}\right) + l & T > T_m \end{cases}$$

Solutions to numerical problems are directly related to the boundary conditions (BCs). The above ground region represents the upper boundary of the system. This region includes the ground surface and the exterior surface of the condenser. For certain types of GCTs (condenser embedded in the soil), it represents the ground surface. Both temperature specified (Dirichlet BCs) and flow gradient (Neumann BCs) have been employed [69,90,94,121]. The temperature boundary has been specified in various ways: as a fixed value [32]; as a time varying function [92]; or even as a value that is modified based on the existing ground temperature [91], as would be the case for modifying an air temperature versus time function in order to apply a ground surface temperature boundary condition. The temperature boundary is determined according to field observation site data or from the yearly air temperature and the resulting ground surface temperature obtained from a nearby meteorological station [55,96,122]. The ground surface energy balance BC was particularly used when designing de-icing and snow melting systems [123]. In this BC, the net heat flux arises from the absorbed solar and the long wave radiation, the sensible and the latent heat transfer between the ground surface and the overlying air with or without the snow cover. Figure 14 presents the geometries and boundaries for a road section with and without GCTs. The 2-D model was employed for the road section without GCTs (Figure 14a), while the 3-D model was applied for the road section with GCTs (Figure 14b). Required values of the convective heat transfer coefficient between ground surface and air, $h_{conv}$, were estimated using the empirical correlations by Duffie and Beckman [124]. Other correlations that can be used are thoseproposed by Givoni and Mostrel [125], Rao [126], and Mc Adams [127]. Palyvos [128] reviewed a large number of convective heat transfer coefficient correlations in linear, power law, and boundary layer form.

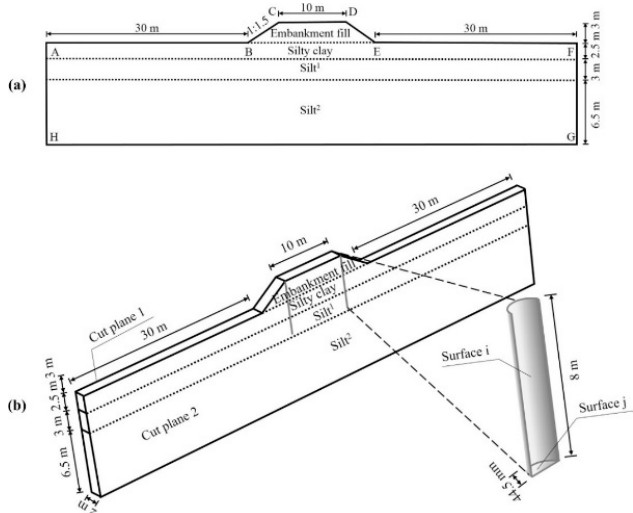

**Figure 14.** 2-D and 3-D model geometries for road sections (**a**) without and (**b**) with GCT [55].

The scientific literature dealing with the modeling of thermosiphon devices contains a number of important studies. An extensive review is available in [129] and in a paper review by Jafari [130]. Briefly, the available literature examines the analytical and numerical approaches, including the lumped capacity model of Dobran [131], the transient thermal network model of Mirzaei [132], the condensation model taking into account the interfacial shear due to mass transfer and interfacial velocity [133,134], and the more sophisticated two- and three-dimensional numerical models [135,136]. However, regarding GTC systems, one major model is applied in almost all publications, and is known as the thermal resistance model. In this model, the performance of the system can be characterized by an overall thermal resistance $R_T$, which is illustrated by a network of thermal resistances and can be calculated by the total sum of the thermal resistances. The thermal circuit between the thermosiphon, the air, and the soil is shown in Figure 15. The model and formulas of the thermal resistances are well-documented in [92,108]. The equivalent thermal resistance model for a two-phase closed thermosiphon is illustrated in Table 4. Some studies (Liu et al. [55]) have simplified the GTC into a line with an approximate heat flow. Others have proposed improved heat transfer models of thermosiphon with or without adiabatic sections and with or without any heat transfer limits [53,97,108,137]. These models can provide a better method for simulating cooling effects of thermosiphons in permafrost engineering.

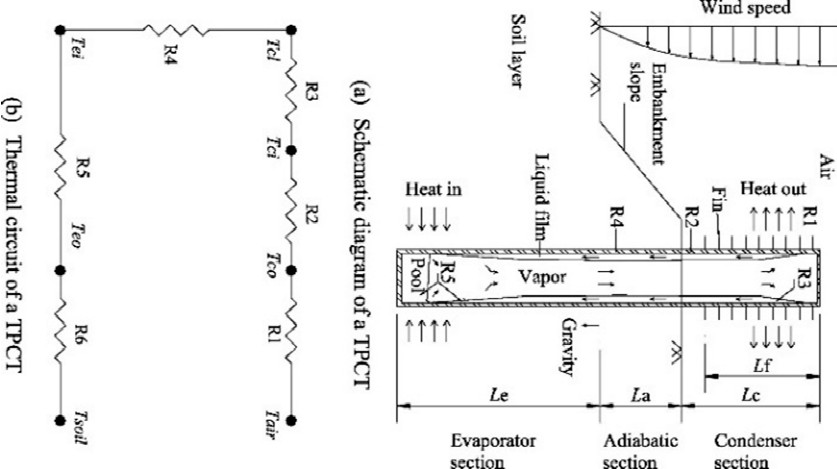

**Figure 15.** Thermal circuit between the thermosiphon, the air, and the soil [92].

A list of correlations is provided in [130] for the calculation of evaporation and condensation heat transfer coefficients.

The performance of GTCs has also been determined using commercially available numerical tools like the Fluent and COMSOL software packages. Indeed, a wide range of software packages has been used. For example, recently, [52,53] used Fluent to model the surrounding soil of a coupled HP and super-long flexible heat pipe. Some examples of computer programs that are commercially available or currently employed by engineers for thermal analysis include, but are not limited to:

- COMSOL software packages [36,91];
- TEMP/W, a commercial finite element model used by [138];
- GEOTHERM, a finite element model developed and used by EBA Engineering Consultants [139];
- THERM2, a finite difference model developed by Nixon Geotech and used by consultants [140];
- Frost 3D [141,142] and Plaxis [143,144]

These numerical tools are useful when identifying critical parameters in design or simulation stages. However, similar to the numerical models previously mentioned, these tools tend to use very fine grids, which necessitate the adoption of a very small mesh and iteration time step. The resulting computation can be time consuming and computationally expensive, especially when a long-term dynamic simulation is required.

Relatively few modeling studies are reported on GCT-assisted HP, GCTs for heat extraction from the ground, or aquifers and HPTs for the production of power and geothermal convectors. The available research has mainly focused on theoretical and experimental thermodynamic methods [1,30]. The latter topic will be discussed later. Most theoretical studies use basic thermodynamic formulations and analysis of the GTC systems. Generally, they attempt to provide relationships for performance in terms of geometric and thermodynamic parameters and discuss the limitations. For example, Ziapour [22] provided an energy and exergy analysis based on the first and second laws of thermodynamics for a Rankine cycle, enhanced by a thermosiphon integrating an impulse turbine.

Regarding thermosiphon-assisted HP modeling, the first approaches ranged from simple thermodynamics models to theoretical calculations. This approach is used generally in order to lead the development of a $CO_2$-GCT-HP system regarding the performance and especially the capacity limit due to critical heat flux, as well as regarding the optimum charge [28,65]. More sophisticated and complete models have been developed over the last few years by [31,32] and [93,99,105,145]. For example, Ebeling et al.'s [31] modelconsists oftwo main parts: heat and mass transfer and fluid flow, which allows the transient heat transfer inside the pipe coupled with heat conduction in the pipe wall and surrounding soil to be modeled. A further elaborated model is the quasi-dynamic model proposed by [32], in which the mass transfer between the liquid and vapor phases, as well as the conduction heat transfer from the surrounding soil towards the pipe, is treated dynamically. However, the film flow modeling is based on the Nusselt theory of film condensation. More details regarding some of these studies are provided in the section on parametric studies.

**Table 4.** Equivalent thermal resistance of a two-phase closed thermosiphon [92,94,95,108].

| Section | Thermal Resistance | Convective Heat Transfer Coefficient ($h$) and Heat Transfer area: $A$ |
|---|---|---|
| Condenser Section | ***R1 + R2 + R3:*** <br> *R1* resistance between the air and the outer wall of the condenser: $R1 = \frac{1}{A_{c,3}h_{oc}}$ <br><br> R2 for the tube wall of the condenser: $R2 = \frac{1}{2\pi \lambda L_c}\ln\left(\frac{d_{oc}}{d_{ic}}\right)$ <br><br> R3 for the liquid film formed inner the condenser: $R3 = \frac{1}{A_{ic}h_{ic}}$ | $h_{oc} = h_{conv}\frac{A_{c,1}+\eta A_{c,2}}{A_{c,3}h_{oc}}$  $A_{c,1} = \pi d_{oc}(L_c - n\delta)$ <br> $A_{c,2} = \pi\left(2n(r_2^2 - r_1^2) + 2n\delta r_2\right)$ <br> $A_{c,3} = \pi d_{oc}L_c$ <br> $h_{conv} = 0:1378\frac{\lambda_{air}}{d_{oc}}Re^{0.718}Pr^{1/3}$ <br> $h_c = 0.0943\left[\frac{\rho_f k_f^3 g(\rho_f - \rho_v)h_{fg}+0.68C_{p,f}(T_v-T_c)}{L_c\mu_f(T_v-T_c)}\right]^{1/4}$ <br> Nusselt theory in [134] |
| Adiabatic Section | $R4 = \begin{cases} 0 & \text{\textit{the thermosiphon is in working state}} \\ +\infty & \text{\textit{the thermosiphon is not in working state}} \end{cases}$ | $A_{ic} = \pi d_{ic}L_c$ |
| Evaporator Section | R5 + R6: <br> R5 for the liquid film and liquid pool in the evaporator: $R5 = \frac{1}{A_{ie}h_{ie}}$ <br><br> R6 for the tube wall in the evaporator: $R6 = \frac{1}{2\pi \lambda L_e}\ln\left(\frac{d_{oc}}{d_{ic}}\right)$ | $A_{ie} = \pi d_{ie}L_e$ <br> $h_e = 0.32\left[\frac{\rho_f^{0.65}k_f^{0.3}C_{p,f}^{0.7}g^{0.2}q^{0.4}}{\rho_v^{0.25}h_{fg}^{0.4}\mu_f^{0.1}}\right]\left(\frac{P_{sat}}{P_{atm}}\right)^{0.3}$ [146,147] |
| Total Thermal Resistance | $R_T = \sum\limits_{i=1,\,6} Ri$ | |

Generally, the numerical and theoretical work reported in the above section, dealing with the thermosiphon device, is supported by experimental information and data for the validation of the predictive models, as well as for confirming some characteristic featuresof thermosiphon and local flow behavior [99]. Several comparisons of simulations and experiments have been carried out and discrepancies were generally small [32,72], so somemodels were able to successfully simulate the performance ofGCTs over the course of 1 h to 50 years, using actual weather data and loads. The major factor limiting the prediction quality of the models was the accuracy of the input data [20]. It may be noted that nearly all validation effortshave been devoted tocomparisons of simulations and experiments, and little attention has been paid to the accuracy of the input parameters to the prediction discrepancies. Recent progress in computational power and CFD developments has contributed to greatly improving the agreement between simulations and experiments. Hantsch and Gross [105] simulated the operation of periodically working geothermal thermosiphons and enabled an analysis of the key factors influencing the performance with remarkable success (less than 0.5% discrepancy on heat flux). Recent comparisons for various applications and conditions have been performed by many authors [31,103], with fairly good predictions for temperatureand efficiency in the respective ranges of 1.0 to 1.4 °C lower than those in the natural ground at the same depths and 2.7–16%.

### 3.1.2. Parametric Studies

The widespread use of GCTs in different applications has led to the implementation of a variety of parametric analyses, based on validated numerical models, to investigate the effect of different parameters on system performance. The objectives and the requirements of each analysis vary from one application to another. Parametric analyses were also undertaken to refine GCT system designs and assist in making a final choice between different systems [52]. Performance has been analyzed in terms of several parameters, such as system thermal stability [148], COP [53], working period of the GCT, and performance of partially wetted GCT [105]. As performance indicators of each application are different and significantly depend on the specific location and application, it was difficult to compare different cases. Many studies have investigated the influence of structure parameters on the working process of GCTs, such as spacing, length, and type of array [53,148,149]. The impacts of parameters like the soil thermal conductivity, thermal diffusivity, and spacing of the heat pipes were investigated for the performance of an array of GCTs [149]. It was found that soil thermal conductivity and spacing between GCTs have significant effects on the temperature rise of the GCTs. The influence of the wetting ratio, tube material (steel and polyamide), borehole radius, and thermal conductivities of the borehole filling (grout) and the soil on the performance of a partially wetted geothermal thermosiphon have been studied by [105]. It was found that the wetting ratio has a strong influence on the performance of polyamide tubing because its thermal conductivity is much lower than the steel. In addition, decreasing the borehole diameter and increasing the thermal conductivity of the filling material is important for enhancing the performance of GCTs. The influence of soil thermal conductivity is of minor importance for short times, whereby at longer times, a reduction of the heat flux for lower conductivities is obvious. Similar conclusions have been also made by [99], which showed that low thermal conductivity pipe and filling materials require a large wetting ratio (for best results, >80%). Furthermore, the filling material should match the thermal conductivity of the surrounding sub-surface to ensure maximum heat transfer to the thermosiphon. The authors concluded that a well-wetted polyamide pipe with a thermal conductivity larger than 2W/m·K could replace the steel pipe.

Various investigations have indicated that the performance of GCT also depends on the tube material, thermal conductivities of the borehole filling and soil, angle inclinations, extraction time, and surrounding environment [99,104,105,150]. In smoothpipes, the falling film tends to rip open when it falls below a certain thickness. The influence of the extraction time on the amount of heat is obvious, since during a long extraction period, the temperature decreases not only close to the pipe, but also in some distances. This leads to a decrease of the heat extraction rate with time until the quasi-steady state is reached. Nevertheless, with the presence of groundwater advection, the variation tendency would

almost level out after several days of decrease [53]. Results of different declining angles of GCTs (0° to 60° with 10° increments) [148] used for the cooling effect and stability of the rail-way embankment showed that when the thermosiphon lied along the slope toe with a declining angle of 25°–30°, the promoted effect of the permafrost table under the embankment of the center, shoulder, and toe was optimal. Besides the inclining angle, there are other influencing factors of the cooling performance of the GCT, including working fluid, aspect ratio (the ratio between evaporator length and internal diameter), and filling ratio (the ratio between the volumes of working fluid and evaporator section). These influencing factors have been extensively investigated by previous researchers ([40,52,53] on the SFHPs). According to Wang et al. [52], several working fluids are selected as possible candidates for SFHPs considering the operating temperature, such as ammonia, pentane, acetone, methanol, propane, ethanol, normal butane, isobutene, and heptane. They found that ammonia was optimal for SFHPs concerning the entrainment limit. They also noted that, due to the high pressure of $CO_2$ up to 45 bar at a temperature of 10 °C, which is too high for a polymer tube, it was not considered in their study.

Further parametric investigations have been conducted in order to understand the heat transfer mechanisms involved in these systems and to improve the computational models for future optimum application designs or to demonstrate the feasibility of the GCT techniques in certain application s [52,53,89,103]. For example, Ma et al. [103] demonstrated that the GCT could be a feasible and effective means of enhancing oil production. In this study, a number of influencing parameters on the efficiency of the thermosiphon were investigated, including the working fluid, the fluid production rate of the oil wells, the water cut of the produced fluid, and the working depth of the thermosiphon. It was reported that under the same working conditions, the wellhead fluid temperature increase using methanol was higher than that using water. The increased water cut and production rates will benefit the effectiveness of the Two-Phase Closed Thermosiphon (TPCT) wells, though the influence of the water cut is lower than that of the production rate. In addition, the effect of the working depth of the thermosiphon on the fluid temperature is relatively limited. Another example presented by Chen et al. [89] explored the impacts of climate warming and aeolian sand accumulation on the performance of thermosiphon, with the thermosiphon being adopted to mitigate thaw settlement of an embankment in a sandy permafrost zone. It was found that the combined effect of climate warming and aeolian sand accumulation largely weakened the cooling effectiveness of the thermosiphon operating in a crushed-rock interlayer embankment (CRIE), accelerated the underlying permafrost degradation, and promoted the development of areas of unfrozen ground surrounded by permafrost in embankment sub-base and sub-grade. Following these impacts, the long-term thermal stability of the CRIE could not be maintained. A further example is the thermal analysis carried out by Wang et al. [52],to investigate the effect of structure dimension, arrangement, material parameters, and climatic conditions on the performance and reliability of an ice and snow melting system (ISMS) using SFHPs heated by shallow geothermal energy. Results revealed the recommended dimensions of SFHP as: $o32 \times 2$ mm in diameter, 40 m evaporator, and 20 m condenser, with a heat output of 1150 W at a steady state under a typical climatic condition. Furthermore, the appropriate filling ratio for SFHPs ranged from 62% to 65%, and the heat throughput of a single SFHP at a steady state was in the range of 850–1200W, varying with climatic conditions.

### 3.2. Experimental Studies on GCTs

Experimental research is a common scientific approach thathas been used in the development of thermosiphons technology for different applications. Two main methods are used in the vast majority of experimental investigations. The first method relies on controlled laboratory experiments in which several test benches (small-scale or full-scale) arebuilt, instrumented, and tested under different controlled operation conditions. In this method, the operator has the ability to adjust or set certain factors at a required level and to record response parameters. The second approach, termed field tests, requires in-situ tests, which are generally full-scale test facilities. Laboratory methods can

make use of steady-state and transient techniques. According to the available literature, the aim of the experimental approach is fourfold, as follows:

- Provide a better understanding of the heat transfer processes occurring in the different GCT types, analyze the cooling and heating mechanisms of the different engineering measures proposed, and evaluate their performance and the main controlling factors;
- Provide laboratory or field data for the validation of numerical simulation models, and the analysis and improvement of long-term performance predictions;
- Further optimize the design, functioning, monitoring, and key design parameters in different environmental settings (e.g., in permafrost regions), and identify the aspects and the methods to be improved;
- Develop new engineering measures for integrating GCTs into the construction to produce a satisfactory performance, technical feasibility, and cost-effectiveness.

It is important to recall that in many instances, the same experimental research can offer valuable information for a better understanding of the heat transfer processes occurring in the GCTs, and key data for model validation and design optimization

### 3.2.1. Performance Evaluation

Numerous experimental works have been published, examining the heat transfer processes occurring in the GCTs, analyzing the cooling or heating mechanisms, and evaluating their performance in terms of the main controlling factors. The earliest in-situ and laboratory experimental works investigated the thermal permafrost stability in the Alaska oil Pipeline [151,152]. Design and performance of foundations stabilized with thermosiphons were reported by [150,153]. In these studies, some areas of research were suggested to improve the design and performance of thermosiphon stabilized foundations, such as: (1) Effect of evaporator installation angle on performance; and (2) detailed comparisons of design versus field performance of thermosiphon stabilized foundations.

In many instances, the performance indicator typically used is the thermal conductance (W/°C) of the thermosiphon, which includes the effects of heat conduction, evaporation, condensation, and radiation and convection within the finned surface. This conductance is defined as the inverse of total thermal resistance of the thermosiphon ($C_T$=$Q$/($T_{soil}$-$T_{air}$)), where Q is the overall heat transfer rate of a thermosiphon (W). Several expressions for the thermal conductance were determined from the tests performed on evaporators of different slopes, charged with $CO_2$ or ammonia [150,152,153]. The thermal conductance was presented as a function of air velocity and the slope of the evaporator section. It was shown that the heat transfer conductance increased with the air velocity and the increasing evaporator slope.

Performance has also been experimentally evaluated in terms of the system working period and heating and cooling capability. The main aspects of cooling performance include the rate of ice growth on the evaporator section [154], the thermal stability of the construction foundation, [54,72,89,96,152], the embankment deformations, the thaw settlement, the temperature distribution of the ground beneath the construction, and the effectiveness of thermosiphons [155]. Overall, laboratory and field observations demonstrated the satisfactory performance of thermosiphon techniques applied to applications such as maintaining permafrost in the subgrade layers of highways and railways. Heating performance has been considered in the context of snow melting systems and GCT assisted HPs. The overall heat extraction rate was generally the main performance indicator of the GCT-HP [29]. Lim et al. [93] presented the performance of a GCT-HP through an annual heating performance indicator. A typical heat extraction rate from the borehole is 40 to 50 W/m [1]. Nevertheless, attention should be paid to the special regional soil conditions. It is worth mentioning that this heat extraction rate is slightly higher than or identical to common pump dependent secondary fluid solutions that extract about 30 to 40 W/m (with common U-pipe 40 × 2.4 mm collectors and flow rates of about 0.5 L/s).

Another interesting indicator of GCT performance is the electric power that can be generated by an HPT [21] or thermosiphon Rankine cycle and the net cycle efficiency [156]. It is important to note that the expected performance of these systems is very low (below 1%) [3]. This is mainly due to the fact that in general, a single pipe is used and the power extraction is mainly related to the kinetic effect. Carotenuto et al. [13] used the heat flow, Q (Watt), transferred by the GTC as a performance characteristic of these systems, which strongly depends on (i) the permeability of the aquifer, (ii) the natural flow of the geothermal liquid, and (iii) the eventuality that the geothermal liquid stagnates in the well.

### 3.2.2. Working Fluid

Several experimental studies have been conducted to evaluate the performance of GCTs with a number of working fluids. GCTs are very sensitive to the working fluid in terms of operation and performance. Thethermos-physical properties of working fluids (heat of evaporation, vapor and liquid density, liquid thermal conductivity, specific heat, and viscosity) generally affect the performance of GCTs. Favourable thermos-physical properties are of primary importance to ensure a high performance in the range of intended use. For example, to enhance the flow return to the evaporator and turbulent flow within the GCTs, the working fluid should have both a high density and low viscosity [157]. To maximize heat transfer, the fluid should have a high latent heat of evaporation and a high liquid thermal conductivity [20]. Increasing the latent heat increases the heat transfer rate for a given mass flow rate. Gas phase fluid properties are not as important because the liquid film controls the heat transfer rate inside the GCT. An appropriate parameter established for a quick comparison of working fluids for a specific temperature range of thermosiphons has been established by means of a dimensional fluid property set in terms of a merit number [19,158]. The selection criteria for the working fluid must also include considerations such as safety issues, chemical compatibility with the piping material, environment preservation, availability, and costs, in addition to the primary performance requirements.

The selection of the working fluid must also be based on thermodynamic considerations that are concerned with the various limitations to heat flow occurring within the GCT. These limitations are discussed in detail in several references [158–160].

Experimental fluid evaluations found in the literature were performed to identify the most appropriate fluid for a chosen working range of conditions. Depending on the application, site conditions, and desired soil temperatures, pressurized working fluids such as R-11, R-12, R-22, R-507, $CO_2$, butane, propane, or ammonia ($NH_3$) have been tested in thermosiphons for geotechnical applications [48]. Earlier studies such as the Trans-Alaska Pipeline adopted ammonia and propane [155,161]. Ammonia is highly toxic, while propane is highly flammable. Working fluids, such as R-123, R-134a, and ammonia, were experimentally investigated for temperature control of a bridge foundation in permafrost regions [52]. The R11 working fluid was used as the geothermal convector, while several types of working fluids were considered for the Heat Pipe Turbine when using geothermal sources of energy [162]. Finally, R-123 was selected as the working fluid because of its high performance efficiency. Ammonia and Freon were utilized as the working fluids for pavement snow melting systems [163]. Isopentane was utilized as the working fluid for GC for geothermal power production [109].

$CO_2$ is non-explosive, non-flammable, moderately toxic, and non-reactive. In addition, from an environmental point of view, it has zero ozone depletion potential and a GWP equal to 1. $CO_2$ was often used for almost all applications since it is less costly to test and has high potential for use as a refrigerant [158]. Today, it is considered one of the preferred working fluids in GCTs. Since most of the heat transfer processes in GCTs occur in a saturation state, the working pressure levels are determined by the ground temperature levels near the borehole. The vapor density of $CO_2$ (at operating pressures of about 30–45 bar) is approximately seven times higher than the density of R-134a, resulting in a high volumetric refrigeration capacity (which leads to small volumetric flow rates and small pressure loss).

Due to a significantly high pressure up to 45 bar at a temperature of 10 °C, carbon dioxide has not been considered for use with polymer tubing.

### 3.2.3. Pipe Material

Not every material is suitable for use with each refrigerant [164]. The selection of the material depends on the chosen refrigerant. As well as the heat transfer characteristics, the ductility of piping material is important [93]. Since a typical length of GCT reaches up to 400 m, the material should resist some deformation. For some applications, a good material load bearing capacity is required. Heuer [20] and Lyazgin et al. [165] pointed out the importance of using appropriate pipe materials that tolerate the pressure levels at which thermosiphons operate and that are suitable from the corrosion point of view (e.g., for long-term contact with groundwater). The fluid should also be chemically stable, non-corrosive, and compatible with the container material of the GCT. This problem currently limits the use of flexible and/or lightweight systems because of the poor chemical stability of materials like plastics or polymer composites. Julia [1] indicated that when choosing ammonia as the working fluid, no copper can be used because of corrosion. Table 5 shows the compatibility of some thermosiphon fluids with pipe material. Lim et al. [93] reported on metallic materials, and pointed out that even though there were several anti-corrosive types, some concerns still remained on corrosion by underground water, organic matter, or even by roots of plants. The cathodic protection system is required to prevent such corrosion. Selection of the pipe material also depends on the ease of fabrication, including weldability, and machineability.

**Table 5.** Some low temperature application fluids with pipe material [166–168].

| Fluid | Casing Materials Compatibility | | | | |
|---|---|---|---|---|---|
| | Metal | | | | Polymers |
| | Aluminum | Cooper | Stainless Steel | Ferritic Steels | PTFE, PCTFE, PVDF, PA, PP |
| Ethane | ✓ | ✓ | ✓ | ✓ | PTFE, PVDF |
| R22 | - | - | ✓ | - | PTFE |
| R410a | - | - | ✓ | - | - |
| Propane | ✓ | ✓ | ✓ | ✓ | ✓ |
| R134a | - | - | ✓ | - | - |
| R11 | ✓ | - | ✓ | ✓ | PTFE |
| Ethanol | acceptable | acceptable | ✓ | acceptable | - |
| Acetone | acceptable | ✓ | ✓ | ✓ | PTFE |
| Ammonia | ✓ | corrosive in presence of moisture | ✓ | ✓ | ✓ |

### 3.2.4. Condenser and Evaporator Design

GCTs with different condenser and evaporator configurations have been tested under controlled laboratory conditions and in situ for different applications. Despite the attractive GCT features of simplicity, reliability, and costs, some configurations have not received sufficient attention compared to others because of the different potential of applications. The simple vertical pipe with a condenser in the air was tested and developed for maintaining permafrost foundations by Erwin Long in 1960 [6]. Extensive laboratory and field tests were conducted to adapt the new technology for the needs of the Trans-Alaska Pipeline project [20]. The thermopile tested in the pipeline was designed to transfer from 40 to 60 W/m. Each thermosiphon was made of 1.5 mild carbon steel, with pipe lengths ranging from 8.5 to 23 m. One characteristic, critical to the operation of vertical GCTs, is that when the GCT is inactive, the liquid pool at its bottom is 0.3 to 0.6 m deep [20]. The vertical thermosiphon has evolved through the years, but the basic principal stays the same. Lyazgin et al. [169] and [170] developed new techniques to stabilize pile foundations against frost heave. These techniques are: (1) a newly developed thermosiphon device, termed TMD-5, to lower the permafrost temperature and increase its strength; (2) new pile cross-section designs such as cruciform piles, H-piles, channel piles, and three-blade piles; and (3) chemical methods. The TMD-5 is manufactured from aluminum alloy with a length from 4 to 11.5m and tube diameter of 28mm. It was found that TMD-5 increased the running time by 1–1.5 months/year. The initial TMD-5 design was later improved to include a new

Thermo-stabilizer with Improved Productivity (TIP) and an enlarged heat-exchange surface. The condenser of the TIP has additional (secondary) finning and the evaporators include a second plate. Figure 16 presents a schematic view of the TMD-5 and TIP designs. The equivalent diameter of TIP increases from 54 to 160 mm. The cooling capability of the TIP is 1.6 times higher than the TMD-5 [171]. More information about TMD-5 and TIP properties, technical parameters, and test results are presented in Bayasan et al. [172,173].

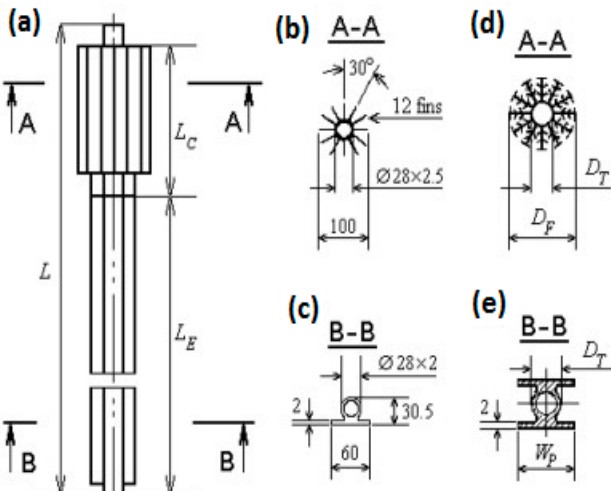

**Figure 16.** View of thermo-stabilizers TMD-5 and TIP (**a**), and cross-sections of: TMD-5 condenser (**b**) and evaporator (**c**), and TIP condenser (**d**) and evaporator (**e**) from [171].

Several other vertical GCT designs used in conjunction with HP include: relatively deep GCTs (150 to 400 m), double tube coaxial, Helix loop, and U-pipe loop. Some details of thermosiphon-assisted HP are covered in paragraph 2.3 of the current paper. The first investigation using $CO_2$ as the working fluid in conjunction with HP was done by Rieberer et al. [174], in which several prototypes, 50–65 m deep and 15 mm in diameter, were tested. A counter current liquid-vapor vertical GCT was presented by [28]. It consists of an18 m deep corrugated stainless steel geothermal thermosiphon, which was experimentally tested and compared to a typical brine system of the same depth. Field tests using distributed temperature measurements along U-pipe and coaxial GCTs installed in groundwater filled boreholes have been performed by Acuña et al. [30] and Julia [1]. Recently, Ebeling et al. [31] and [32] reported their experimental results on two geothermal GCTs drilled about 400 m deep for a heating application. One is a plain steel pipe with an inner diameter of 114 mm and the other is a helically corrugated pipe with an inner diameter of 98 mm. $CO_2$ is used as the working fluid in the TPCT. The total filling mass of $CO_2$ is 642 kg.

Sloping evaporators are installed on the majority of existing passive subgrade cooling system installations. These systems often do not require the same evaporator depths as conventional vertical configurations, significantly reducing project costs. Typically, evaporator slopes are between 10% and 3%, with the median being 5%. Observations over a five-year period of the performance of the first sloping evaporator were presented for a school constructed at Ross River, Yukon Territory, in 1975 [51]. The evaporator sections were made of aluminum tubes that extended into a gravel pad supporting a slab-on-grade foundation. It is worth mentioning that the laboratory tests were also conducted by [153] on two full size thermosiphons. They presented heat transfer conductance for air velocities ranging from 0 to 5.2 m/s and evaporator angles varied from 0 to 12°, from the horizontal. It was found that increasing air velocity and increasing evaporator slope angle increased the heat transfer conductance. This agrees with the laboratory studies done by Lee and Bedrossian [175] and Negishi and Sawada [176], who conducted tests with small thermosiphons at inclined angles. Similar results have also been found in field tests conducted by Zarling and Haynes [177].

For many applications, it is desirable to install the evaporator of such a system horizontally. Such applications include large buildings, aircraft hangars, or roads and railroads. Horizontal thermosiphons were first tested by Denhartog [178] at a laboratory scale. A patent application was submitted for this idea by [179], with the purpose beingto develop a thermosiphon that is able to function effectively, with the elongated evaporator having a positive or negative slope. Sixteen single evaporator units with vertical finned condenser sections were installed and tested in the Chena Hot Springs Road project in 1998 [180]. The evaporator pipes consisted of a 65 mm nominal diameter and a length of either 12.8 m or 14.0 m. They were connected to an 80 mm nominal diameter, 2.4 m long-finned condenser section with an area of 6.5 m$^2$. As mentioned above, laboratory tests on small scale thermosiphons performed by Lee and Bedrossian [175] and Negishi and Sawada [176], showed a significant decrease in the heat transfer conductance with a horizontal evaporator. Laboratory tests were also conducted on a full–size thermosiphon with a horizontal evaporator that was 37-mlong and 7.6 cm in diameter [154]. The performance was evaluated by measuring the rate of ice growth on the evaporator, which provides an indication of the heat transfer conductance of the thermosiphon. System conductance values were determined for wind speeds of 0 to 5.4 m/s applied to the condenser section. Haynes et al. [154] indicated that as the thermal conductivity of frozen soil is about 20% higher than that of ice, the performance values obtained in this study are conservative for a field design application.

A GCT with a flat loop evaporator was first field tested in south Winnipeg, Canada, in 1994 [11]. Its performance was compared to sloped evaporator GCT in Winnipeg during the winter of 1993–1994. Typically, tested flat evaporators are manufactured using a 20 mm steel pipe. It was found, after a winter of operation, that the flat evaporator unit froze 1.4 times more soil volume than the sloped evaporator unit. Twelve flat evaporators units were installed and tested in Chena Hot Springs Road [180]. This was part of a demonstration project of three new types of thermosiphons, including vertical, flat loop, and buried. The evaporator piping for these units has a 20 mm outside diameter forming horizontal grids that are 2.7 m wide by 12.2 m or 14.2 m long. They are connected to 80 mm nominal diameter finned vertical condenser sections of a 4.9 m nominal length with an area of 15.8 m$^2$. Two years of ground temperature measurements have been analyzed for the test site. The authors reported that vertical and flat evaporator units cool the ground in a very similar manner. However, because flat evaporators units have a larger fin area, this makes them more effective than the vertical evaporators units. A new technique has been developed to release the heat picked up from the subgrade in the near surface soils or embankment. This technique makes use of a buried GCT beneath the ground surface, providing an opportunity for road and runway installations that result in no surface safety hazards [8]. Experimental works done on buried and hairpin GCTs were published by [54,180]. Probes were installed in the ground beneath the embankments and on the condenser and evaporator pipes of one of the hairpin thermosiphons. The performance during the first year of operation was examined by analyzing evaporator and condenser temperatures and heat flux values. For interested readers, these types of GCTs are covered in more detail in [8,11,24,181].

Another aspect that was experimentally investigated by [152,153], is the most favourable fined surface dimensions under which the system could be used more efficiently. The performance of the finned section has a major impact on the performance of the system because the air side heat transfer coefficient on the finned section is 10 to 50 times smaller than the condensing and the boiling coefficients. The main parameters affecting the air side heat transfer coefficient are fin spacing and efficiency, configuration, and radiative properties of the fin coating. Note that the work presented by [153] is an extension of the work done by [152]. Two full size commercial thermosiphons were experimentally investigated in the study. The first one was filled with $CO_2$ and the second with ammonia. Two types of fins have been tested (Figure 17): steel annular fins of rectangular sections (horizontal fins) and aluminum straight fins of rectangular sections (vertical fins). More details on the fin assembly, dimensions, and the corresponding condenser sections are presented in these studies. Zarling [152] indicated that the $CO_2$-filled system with horizontal fins offered a superior performance under windy conditions. However, the ammonia filled system with vertical fins had a higher performance under

still air. This was explained in [153] by the fact that long vertical fins on the ammonia system promote natural convection associated with a low wind speed, while the segmented annular fins on the $CO_2$ unit helped with forced convection associated with high wind speeds. In addition, the effect of the thermosiphon nearby building walls was also experimentally determined in the latter study. It was found that the heat transfer for the thermosiphon with the segmented condenser fins was relatively insensitive to the location and proximity of the building walls. The heat transfer for the thermosiphon with the longitudinal fins was slightly reduced by the walls at lower air velocities.

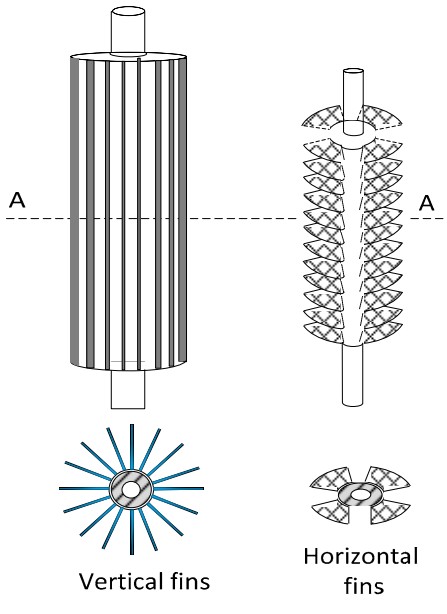

**Figure 17.** Condenser fins.

### 3.2.5. Experimental Data

Providing laboratory or field data is an expensive and time-consuming task, which generally needs a lot of provisions (efforts) in terms of equipment design, instrumentation calibration, and specific knowledge of measuring techniques and test benches and prototype building. Experimental plans are often established in advance, and collected data is used for model validations, laying a solid basis for proper model development, system design, successful construction, and safe operation of the system. For validation purposes of GCTs, transient or steady state global parameter measurements such as air and ground temperatures, pressures, wind velocity, ground properties (soil thermal conductivity, water content, and density of the soil), and mass flow rates are collected for specific conditions at strategic locations [13,32]. Collected data are used directly or indirectly for the calculation of certain parameters and indicators of performance of the GCTs systems. Given the variety of analytical and numerical models used for simulation and analysis system performance, validation requirements are also varied. Some GCT models have been validated by comparison with experiments [53,69,108] making use of these global data and associated operational information. For example, Zhang et al. [108] compared calculated and measured outer-wall temperatures at the evaporator section of GCT at left and right embankment shoulders after one year of the construction. Other studies compared spatial distributions of simulated temperature at different depths of the GCT [98,99] and in the ground around the GCT [34,92,94,96,98,182]. In many cases, field data and associated operational information have been used as initial and boundary conditions for numerical models [72,89,96]. Recently, Ebeling et al. [32] reported their experimental results on two geothermal $CO_2$ thermosiphons drilled about 400 m deep underground, which they compared to simulations in steady state. In addition, to validate the results of the dynamic and quasi-dynamic simulation models, two sets of field measurement data were

used from a 400m vertical $CO_2$ GCT [31,32,145]. Combined global temperature (flow rate and pool temperature) and the spatial temperature distribution along the pipe were used in the validation. The numerical results were compared with experimental data and a good agreement with the measured data for a long-term operation was achieved. However, the start-up effects are not yet represented very precisely to be adequately accounted for.

It is worth mentioning that, in addition to these global and distributed field data, other techniques have been employed, ranging from visual infrared scanning, fiber-optic cable, and visual observations to the measurement of internal pressures [60,99,150,183–186]. Direct visual observation of the liquid film flow in GCTs in some cases may be necessary to confirm certain theoretical model predictions, or to update some existing theories. Storch et al. [185] utilised a test facility for visual observations of the falling liquid film flow and its distribution around the inner tube surface by the application of a specially designed camera system, introduced inside the pipe, down to its lower end. Hartmann [99] expressed an equation of the wetting ratio, which was measured by visual observations along the entire length of the pipe. Zorn et al. [60] used a monitoring system, consisting of a fiber-optic cable, a weather station, and an infrared camera system installed at a site. This monitoring system was used to validate their theoretical calculations. Zarling et al. [150] employed an infrared thermal camera to monitor the condenser.

### 3.2.6. Optimized Design, Operation, and Monitoring

Development of the GCTs has been driven by the fabrication and testing of various prototypes. Development has typically aimed to simplify manufacture, finalize design, and/or optimize the performance by successive prototype improvements [20,24,30,187,188]. While developing designs, emphasis has generally been placed on reliable short- and long-term operation. Selected commercially available software means were calibrated with existing performance records to identify optimized designs, which were generally adapted to specific application environments. Changes due to ground geology, climate warming, thermal influence, and load of the installation can only be identified through testing. Therefore, various field tests have been performed to develop and optimize the design for short- and long-term use [1,11,24,30,188]. For example, Holubec [24] presented a study to determine the causes of poorly functioning flat loop thermosiphon foundation projects, and to assess the suitability of this design as a foundation system for a 50-year life span of a building on warm permafrost, subjected to climate warming. It was found that poor functioning of the few buildings supported by thermosiphon foundation was related to: a) poor design/construction of the granular pads on which the thermosiphon evaporator pipes are founded, b) inadequate construction details and construction scheduling, and c) inadequate insulation design. Julia [1] showed that no natural circulation was achieved for the first-iteration design of a thermosiphon loop for heat extraction from the ground, composed of a central insulated pipe and a return line in spiral form.

Monitoring systems have been installed to detect poorly performing systems and major malfunctions. As indicated above, a variety of techniques range from visual infrared scanning to the measurement of internal pressures. Detailed information about these monitoring techniques was well-summarized by Wagner [8]. For example, visual observations of falling film flow inside a GCT was performed by Storch et al. [186]. A test setup with a special pressure lock system for the heat pipe's head was developed, whereby a miniature camera could enter the heat pipe's section. For further details on this important topic, the reader may also refer to the dedicated paper by Yarmak and Long [183]. The authors of this paper present a comprehensive review of the monitoring techniques currently in use and make recommendations to establish monitoring plans for thermosiphons. To the best of the authors' knowledge, the maximum monitoring time was carried out on a thermosiphon roadway test site spanning for 11 years [189]. Eight years of monitoring data of thermosiphon embankment were collected at the QTH [190]. A five-year set of ground temperatures and embankment deformations at different soil layers was also monitored by Yu et al. [94] at a road section with GCT and at a neighboring road section without GCT. McFadden [184] reported his work evaporator and ground

temperature temperatures covering a four-year span of operation from 1982 to 1987. One example of the anon-conventional monitoring method is the infrared system mounted in a helicopter by Heuer [20]. Data was registered on video tape, displayed on a standard TV screen, and then analyzed. This system was found to be the best method of monitoring GCTs radiator temperatures. It is also worth mentioning the variety of monitoring tools used by Lynn and Rhodes [75] to evaluate the effectiveness of a vertical frozen soil barrier at Oak Ridge National Laboratory. Monitoring tools used including groundwater dye tracing, groundwater measurements, and subsurface soil temperature. Data collected during the demonstration provided evidence that the frozen soil barrier was effective at isolating the pond.

### 3.2.7. Remarkable Engineering Features of Thermosiphons

Different engineering measures have been taken and experimentally investigated in order to guarantee continuous GCTs operation. It was, for example, vital to analyze new solutions or techniques for new GCT applications, and confirm experimentally that these systems always remain within their limits of the operating conditions (e.g., heat load, ambient temperature, and orientation) and the chosen geometry. New engineering measures may consist of new component designs, measurement techniques, pipe materials, or working fluids. To cite only a few, Zhi et al. [55] suggested the combination of thermal insulation and GCT to protect the warm permafrost. Ma et al. [161] proposed remedying embankment thaw settlement in a warm permafrost region with thermosiphons and crushed rock revetment. Lai et al. [191] presented an L-shaped thermosiphon with crushed-rock revetment and insulation and carried out a laboratory test to investigate the joint cooling effect of this new kind of embankment. Furthermore, innovations have been made in shapes of the thermosiphon, for example, the hairpin thermosiphon proposed by Jianfeng et al. [54]. Regarding GCT-HP, as indicated previously, special $CO_2$ condenser designs have been developed and even patented [28], in order to ensure a smooth film at the inner tube wall condenser.

The reliability of the GCT was experimentally demonstrated in several applications. McKenna and Biggar [192] studied the rehabilitation of a passive-ventilated slab on grade foundation using horizontal thermosiphons. After three years of temperature data recording beneath the slab, the authors showed that the horizontal thermosiphon was operating as planned, by reducing maximum temperatures in the ice-rich native soil sufficiently below zero to stop the thawing process.

The subject of the feasibility of GCT technology has been experimentally demonstrated for many applications, as well as many fits, including its environmentally friendly nature and low required maintenance. Many demonstration projects have been carried out to prove the feasibility of extracting geothermal energy by means of GCTs [156,162,193]. Franco and Vaccaro [3] indicated that the feasibility of the geothermal project becomes more important when the enthalpy of the source considered is relatively low. Typical issues are excessive fluid extraction rates, poorly conceived reinjection strategies, and scaling. The feasibility of GCT-HP systems was demonstrated in various laboratory tests and field investigations by Wang et al. [53] and Kruse [28]. The technical feasibility of a snow-melting system using shallow geothermal energy, especially a system that employs low-temperature heating fluid, has been addressed by [60]. Wagner and Yarmak [17] also demonstrated how quickly a frozen barrier can be created by hybrid thermosiphons. Kusaba et al. [162] showed that GCT technologies have the potential to utilize heat from vacant wells and generate electric power, with the authors suggesting that that it could possible to generate 7.8kW of electric power for every100 kW of heat extracted. A demonstration system for geothermal heat extraction using large size GCT, 150 mm OD and 150 m long, with showering nozzles, is presented in [188]. The system is able to extract 90 kW heat continuously with 3000 W/m$^2$ heat flux at the evaporator. Finally, several works proved the feasibility of GCTs for the thermal stabilization of soils (including freezing of thawed soils and cooling of permafrost) and for providing stable support for buildings and structures in cold regions [7,98,191,194,195].

## 4. Conclusion

The available literature on GCT studies is characterized by significant inconsistencies and differences in nomenclature, underlying theories, and methodologies. These incompatibilities stem in

part from the diverse applications and backgrounds of the researchers and the practitioners in these fields. Looking at this technology from different angles of interest, this paper provides a review based on the current published literature on the different types of existing ground coupled thermosiphons, used in a variety of applications requiring moderate and low temperatures. The most important items distinguishing the present paper from the few review papers available in this area are fourfold: (1) Effort has focused more specifically on their classification according to type, configurations, major designs, and chronological year of publication rather than the general system definition. Several summaries are provided in the form of tables gathering important technological findings and characteristics; (2) advances are identified in terms of the latest device developments and innovative concepts of thermosiphon technology used for the transfer of heat to and from the soil; (3) different applications are presented in a novel, well-defined classification in which major GCT applications are categorized in terms of medium (power generation) and low temperature technologies; and (4) performance evaluation.

For each of these topics, the available literature contributions are gathered to handle particular issues. Relevant aspects discussed are related to performance indicators and modeling approaches.

The main concluding elements emerging from the present review may be stated as follows:

- The inclusion of GCT technology has been successfully demonstrated in different studies for a variety of applications. Further cases continue to be under study either because they are still at the concept stage (large scale extraction of geothermal energy and power production using GCTs) or they are part of newly undertaken research (e. g, smart thermosiphons);
- A variety of analytical and numerical models have been used to evaluate and improve the performance of GCTs. The objectives and the requirements of each development case vary from one application to another. As performance indicators of each application are different and depend significantly on specific location and application characteristics, it was difficult to compare different cases;
- The development of the GCT-HP concept appears to be very interesting not only for single-family house uses, but also for meeting the needs of high capacity urban heating systems. However, their use remains limited compared to conventional ground source heat pumps;
- Given increased concern over the impacts of global warming, the "hybrid GCT" approach may represent the best approach for designing important infrastructures in permafrost. This approach moves from passive to proactive permafrost cooling, in order to better deal with the potential consequences of global warming.

**Funding:** This research was funded by Office of Energy Research and Development (OERD) of Canada throughout Program of Energy Research and Development (PERD), BE3-16. The authors would like to acknowledge Justin Tamasauskas, Research Officer at CanmetENERGY-Varennes, for reviewing this paper.

**Conflicts of Interest:** The authors declare no conflict of interest.

## Abbreviations

| | |
|---|---|
| A | area ($m^2$) |
| C | conductance (m/°C) |
| cp | specific heat J/(kg·°C) |
| d | diameter (m) |
| g | acceleration of gravity ($m/s^2$) |
| R | thermal resistance (°C/m) |
| H | enthalpy per unit mass |
| h | convection heat transfer coefficient ($W/m^2$·°C) |
| k | thermal conductivity W/(m·°C) |
| L | length of section |
| $\dot{m}$ | mass flow rate (kg/s) |
| P | pressure (Pa) |

| | |
|---|---|
| Q | heat flow (W) |
| q | heat flux ($W/m^2$) |
| r | radius (m) |
| T | temperature (°C) |
| t | time (s) |
| x | vapor quality (dimensionless) |
| COP | coefficient of performance |
| CL-GCT | closed loop, ground coupled thermosiphon |
| GCT-HP | GCT-assisted Heat Pump |
| GCHP | ground coupled heat pumps |
| GCT | ground coupled thermosiphons |
| GC | geothermal convector |
| HPT | heat pipe turbine |
| PA | polyamide |
| PCTFE | polychlorotrifluoroethylene |
| PP | polypropylene |
| PTFE | polytetrafluoroethylene |
| PVDF | polyvinylidene fluoride |
| ST-GCT | single tube, ground coupled thermosiphon thermosiphon |
| TMD | thermo-stabilizer with Improved Productivity |
| TRC | thermosiphon rankine cycle |
| TPCT | two-phase closed thermosiphon |
| VDT | vapordynamic thermosiphon |
| $\alpha$ | thermal diffusivity ($m^2/s$) |
| $\Delta$ | gradient |
| $\eta$ | heat exchange efficiency |
| $\mu$ | dynamic viscosity (Pa.s) |
| $\varphi$ | diameter (m) |
| $\rho$ | density ($kg/m^3$) |
| atm | atmospheric |
| c | condenser section |
| conv | convection |
| e | evaporator section |
| eff | effective |
| f | fluid |
| i | inner pipe wall |
| l | liquid film |
| m | mean |
| o | outer pipe wall |
| s | soil/solid/freezing |
| sat | saturation |
| T | total |
| v | vapor |
| w | water |

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
