# Peer review of "Ground-Coupled Natural Circulating Devices (Thermosiphons): A Review of Modeling, Experimental and Development Studies"

_inventions, doi:10.3390/inventions4010014_

Reviewer 1 Report

The article describes a review of modelling, experimental and development studies of ground-coupled natural circulating devices (thermosyphon). These devices are very important in engineering and a review of the current literature on the different types of existing ground coupled thermosyphons for use in applications requiring moderate and low temperatures is very useful. 

Author Response

The authors would like to thank the editor and the reviewers for the time they devoted to our contribution and for the useful comments they made to improve its content. In this response, we have made all the efforts to address as thoroughly as possible all reviewers concerns, as detailed below. We hope that our revision has improved the paper to a level of the editor and the reviewers’ satisfaction.

Reviewer 2 Report

The manuscript gives a review on the ground-coupled thermosiphons (GCT) technology by presenting the classification, development history, latest developments, applications, related simulation results and experimental studies. The authors reviewed a large number of literatures that can cover most research of GCT. However, the impact of this review paper can be further enhanced. The manuscript can be considered for acceptance if the authors can well address the following comments:

1)      Please improve the English writing. Some wordings in the manuscript should be modified to be more professional, for example, it should be “wickless” instead of “no wick”. Furthermore, it’s better to use “gravitational and non-gravitational” rather than “wick and wickless”. Some typos are found in the manuscript, such as on Page 31, the Greek symbols, on the second line, I doubt that it should be “heat exchange efficiency”. Please check.

2)      For the simulation and experimental studies part, the authors just presented all the results from the literatures, it is better to provide some discussion. For example, whether there is a good agreement between the simulation and experimental results. Some deep discussion should be provided for a good review paper.

3)      For the wickless loop GCT, there is also a loop system called looped parallel thermosiphon (LPT) system which is combined by multiple loop tubes. Although the underlying principle is not much different from the single tube, it can involve many unique studies such as structure design, parametric studies and engineering application challenges for LPT. As the current manuscript would like to achieve a meticulous and complete review, the authors should also consider this part.

4)      For the applications part, the classification the authors presented is a bit strange. For the power production, when we need to produce some power, we need to extract it first, however the authors choose to show them in a parallel way. Please clarify.  

5)      In the Nomenclature, some units are missing, such as the length of section.

6)      Please provide the full name when using abbreviation for the first time. For instance, on page 2, the ST-GCT and CL-GCT.

7)      Wordings to describe to same object should be consistent. For example, in figure 1, the authors use “thermosyphon”, but most parts of the manuscript use “thermosiphon”. Please correct.

8)      Figure caption format should be consistent. In the figures 2 and 3, the authors utilize the “(a), (b)” after the schematic name, but in figure 6, the authors put them before the name. Please check the format in the whole manuscript carefully.

Author Response

Response to editor

The authors would like to thank the editor and the reviewers for the time they devoted to our contribution and for the useful comments they made to improve its content. In this response, we have made all the efforts to address as thoroughly as possible all reviewers concerns, as detailed below.We hope that our revision has improved the paper to a level of the editor and the reviewers’ satisfaction.

Response to commentsof the reviewer II:

The manuscript gives a review on the ground-coupled thermosiphons (GCT) technology by presenting the classification, development history, latest developments, applications, related simulation results and experimental studies. The authors reviewed a large number of literatures that can cover most research of GCT. However, the impact of this review paper can be further enhanced. The manuscript can be considered for acceptance if the authors can well address the following comments:

1)   Please improve the English writing. Some wordings in the manuscript should be modified to be more professional, for example, it should be “wickless” instead of “no wick”. Furthermore, it’s better to use “gravitational and non-gravitational” rather than “wick and wickless”. Some typos are found in the manuscript, such as on Page 31, the Greek symbols, on the second line, I doubt that it should be “heat exchange efficiency”. Please check.

Response to comment 1:

To address the reviewer concern about ‘’English writing’’ and typos, improvements in language have been added to the manuscript (modifications are in red or blue colors). Modifications have been also brought to Figure 1, 2, and 3. The Authors hope that this improvement comply with the reviewer’s comment.

Note that the authors have decided to use the terms “wick and wickless” rather than “gravitational and non-gravitational”, since they consider they are more appropriate.

2)   For the simulation and experimental studies part, the authors just presented all the results from the literature, it is better to provide some discussion. For example, whether there is a good agreement between the simulation and experimental results. Some deep discussion should be provided for a good review paper.

 Response to comment 2:

To account for this comment, 2 paragraphs were added to the manuscript discussing the agreement between the simulation and experimental results. These paragraphs are:

Paragraphs 1 from line 360 to line 391

Paragraphs 2 from line 502 to line 518

3)     For the wickless loop GCT, there is also a loop system called looped parallel thermosiphon (LPT) system which is combined by multiple loop tubes. Although the underlying principle is not much different from the single tube, it can involve many unique studies such as structure design, parametric studies and engineering application challenges for LPT. As the current manuscript would like to achieve a meticulous and complete review, the authors should also consider this part.

Response to comment 3:

We strongly agree with the reviewer that LPTs can involve many unique studies such as structure design, parametric studies and engineering application challenges. However, we gently remind him that our review deals only with ground coupled thermosiphons for use in applications requiring moderate and low temperatures. We have not found in the currently published literature any work on LPT in geothermal applications.

4)     For the applications part, the classification the authors presented is a bit strange. For the power production, when we need to produce some power, we need to extract it first, however the authors choose to show them in a parallel way. Please clarify.  

In this figure authors want to show major applications for the exploitation of medium and low temperature geothermal sources. For medium temperature applications authors showed 2 things: 1) the use of this kind of energy (Large scale heat extraction for direct use, power production), 2) potential devises reported in the literature to use this source of energy.

Note, modifications have been also brought to Figure 12 and to the text explaining this figure.

5)      In the Nomenclature, some units are missing, such as the length of section.

 Response to comment 5:

Missing units are added to the manuscript.

6)      Please provide the full name when using abbreviation for the first time. For instance, on page 2, the ST-GCT and CL-GCT.

Response to comment 5:

Full name of abbreviations is provided in the manuscript.

7)      Wordings to describe to same object should be consistent. For example, in figure 1, the authors use “thermosyphon”, but most parts of the manuscript use “thermosiphon”. Please correct.

Response to comment 7:

Thermosiphon term is used and modification have been made in the manuscript.

8)      Figure caption format should be consistent. In the figures 2 and 3, the authors utilize the “(a), (b)” after the schematic name, but in figure 6, the authors put them before the name. Please check the format in the whole manuscript carefully.

Response to comment 8:

Figure caption format is checked and only one format is adopted in the whole manuscript.

Round  2

Reviewer 2 Report

The authors have well addressed the comments. However, it is better that the authors add a few more recently published paper about thermosiphons. See below:

Panse, S. S., & Kandlikar, S. G. (2017). A thermosiphon loop for high heat flux removal using flow boiling of ethanol in OMM with taper. International Journal of Heat and Mass Transfer106, 546-557.

Traipattanakul, B., Tso, C. Y., & Chao, C. Y. (2019). A phase-change thermal diode using electrostatic-induced coalescing-jumping droplets. International Journal of Heat and Mass Transfer135, 294-304.